

# Test-Time Scaling with World Models
# for Spatial Reasoning

**Yuncong Yang**[1][*]**, Jiageng Liu**[1][*]**, Zheyuan Zhang**[2]**, Siyuan Zhou**[3]**,**
**Reuben Tan**[4]**, Jianwei Yang**[4][†]**, Yilun Du**[5]**, Chuang Gan**[1]
[1]UMass Amherst, [2]JHU, [3]HKUST, [4]Microsoft Research, [5]Harvard
yuncongyang@umass.edu

https://umass-embodied-agi.github.io/MindJourney/

## Abstract

Spatial reasoning in 3D space is central to human cognition and indispensable for embodied tasks such as navigation and manipulation. However, state-of-the-art vision–language models (VLMs) struggle frequently with tasks as simple as anticipating how a scene will look after an egocentric motion: they perceive 2D images but lack an internal model of 3D dynamics. We therefore propose MindJourney, a test-time scaling framework that grants a VLM with this missing capability by coupling it to a controllable world model based on video diffusion. The VLM iteratively sketches a concise camera trajectory, while the world model synthesizes the corresponding view at each step. The VLM then reasons over this multi-view evidence gathered during the interactive exploration. Without any fine-tuning, our MindJourney achieves an average 7.7% performance boost on the representative spatial reasoning benchmark SAT, showing that pairing VLMs with world models for test-time scaling offers a simple, plug-and-play route to robust 3D reasoning. Meanwhile, our method also improves upon the test-time inference VLMs trained through reinforcement learning, which demonstrates the potential of our method that utilizes world models for test-time scaling.

## 1 Introduction

Humans inhabit—and effortlessly reason about—a three-dimensional world. Our innate 3D spatial understanding allows us to plan routes, manipulate objects, and make decisions in cluttered environments. Spatial intelligence is a fundamental component of human cognition, developing progressively throughout early life [Gardner, 2011, Vasilyeva and Lourenco, 2012, Tommasi and Laeng, 2012, Lohman, 2013, Moore and Johnson, 2020, Lauer and Lourenco, 2016]. As such, reasoning underpins almost every physical task, and endowing embodied agents with comparable 3D intelligence has become a central goal of embodied AI research [Koh et al., 2021, 2023, Yang et al., 2023, Wang et al., 2024b, Du et al., 2024, Zhu et al., 2025a]. Recent advances in vision–language models (VLMs) have produced systems that are already comparable to human performance in visual recognition and comprehension, even solving simple spatial-reasoning problems [Chen et al., 2024a, Zhang et al., 2025b]. However, these models still fall short of reasoning about the actual 3D world that lies behind a 2D image as humans do. Recent spatial reasoning benchmarks [Yang et al., 2024, Ray et al., 2025, Zhang et al., 2025b,a] reveal that VLMs struggle on tasks that require imagining the consequences of egocentric movements. Given a question shown in the upper-left corner of Fig. 1, a state-of-the-art

---

[*] Equal Contribution
[†] Work done while at Microsoft Research

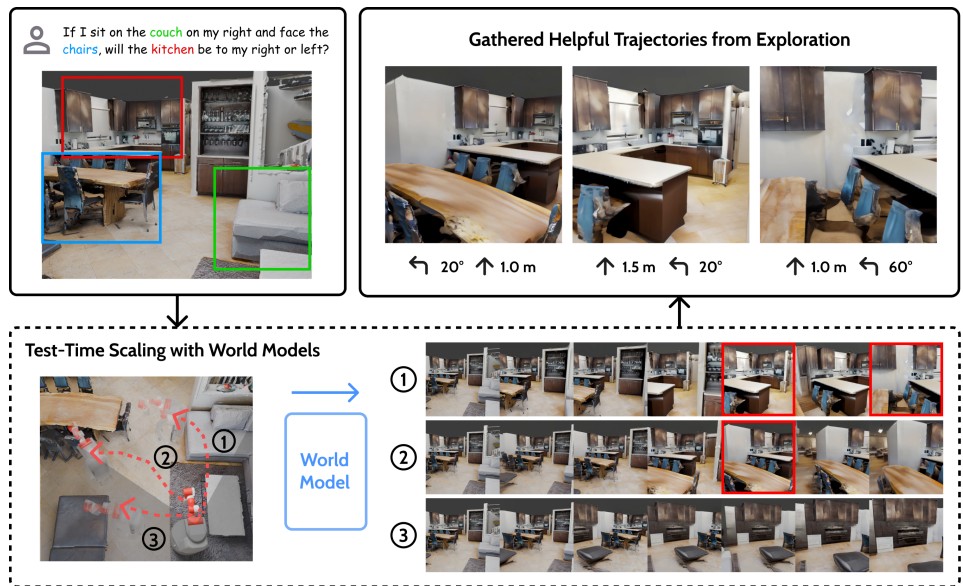

Figure 1: **MindJourney.** Given a spatial reasoning query, our method searches through the imagined 3D space through a world model and improves VLM's spatial understanding through generated observations during test-time.

VLM easily fails on such a perspective-shift question, while given the same view, humans can solve it effortlessly by mentally simulating a short walk and visualizing the next view. This gap highlights a critical bottleneck: current VLMs do not yet treat an image as an interactive world, limiting their usefulness for embodied agents operating in 3D space.

Encouragingly, recent controllable video diffusion models hint at a solution. Systems such as CamCtrl3D [Popov et al., 2025], CameraCtrl [He et al., 2024], Stable-Virtual-Camera [Zhou et al., 2025a], and the History-Guided Transformer [Song et al., 2025] take a single RGB frame plus a pose trajectory and synthesize a coherent, egocentric video that faithfully follows the specified motion. In effect, they serve as world models: given "where the agent will go", they imagine "what the camera will see".

Leveraging this capability, we introduce MindJourney, a test-time scaling framework that enables a VLM to reason by interactively searching and exploring the imagined 3D space through a world model. Confronted with a spatial-reasoning question, MindJourney does not ask the VLM to answer immediately. Instead, the VLM iteratively plans a short exploratory trajectory, with the world model rendering the corresponding egocentric video at each step. The VLM then uses its gathered imagined rollouts in the world to reason and answer the spatial-reasoning question. As shown in Fig. 1, the world model helps scale up the observation space given the image provided in the question, and helpful trajectories with descriptions and observations are gathered for the VLM to answer the question easily. By combining the high-level reasoning ability of a VLM with detailed 3D scene understanding of a world model, we are able to boost the VLM's spatial-reasoning performance significantly.

In a comprehensive evaluation on the SAT benchmark, SpatialNavigator achieves substantial gains on multiple spatial reasoning tasks. Our method improves top-1 accuracy performance across four very different VLM back-ends with two distinct world models by an average 7.7%, with the largest single gain over 10%. Our method also outperforms and can improve upon the o1-like test-time scaling VLMs, which demonstrates the compatibility of our method and showcases the potential of the idea of test-time scaling with world models.

In summary, our key contributions are as follows:

- We introduce MindJourney, the first test-time scaling framework that couples a VLM with a controllable video world model, enabling searching through imagined 3D space to improve 3D spatial reasoning without finetuning.
- We empirically demonstrate that our model offers significant improvement on multiple spatial reasoning tasks.
- We demonstrate that our method is model-agnostic: it boosts four different VLMs with two distinct world models.

## 2 Related work

**World Models** Video generation models have shown their potential as promising world models for gaming and robotic applications [Yang et al., 2023, Bar et al., 2024, Bruce et al., 2024, Gao et al., 2024]. Early works [Du et al., 2023, Bruce et al., 2024, Zhou et al., 2024a, Du et al., 2024] leverage strong video generation models to imagine the future frames for decision making. However, previous works primarily focus on static cameras and simple environments. More recently, several studies [Bar et al., 2024, Parker-Holder et al., 2024, Zhou et al., 2025b, Team et al., 2025] have explored the simulation of 3D environment dynamics using controllable video models that are conditioned on the actions or camera movement, thereby enhancing the spatial imagination capabilities of world models. In our work, we leverage the imagination abilities of world models to help the spatial reasoning ability of vision-language models.

**Vision-Language Models and Spatial Reasoning** The growing advancements in vision-language models (VLMs) have rapidly progressed on various downstream tasks [Radford et al., 2021, Li et al., 2022a, Alayrac et al., 2022, Achiam et al., 2023, Team et al., 2023, Driess et al., 2023]. Many strong open-sourced models have been developed from visual instruction tuning on paired text-image data [Dai et al., 2023, Liu et al., 2023b, 2024, Dong et al., 2024, Yao et al., 2024]. More recent works explored region-level and pixel-level grounding [Li et al., 2022b, Ma et al., 2023, Wang et al., 2024c, Rasheed et al., 2024, Zhang et al., 2024], more training strategies, and test-time scaling [Chen et al., 2024d,c, Wang et al., 2024a, Dong et al., 2025, Zhu et al., 2025b]. Recently, spatial intelligence has gained attention in the VLM community [Liu et al., 2023a, Kamath et al., 2023]. However, evaluation benchmarks SpatialRGPT, SAT, COMFORT, SPAR show that state-of-the-art VLMs still fall short on spatial understanding and reasoning [Cheng et al., 2024, Ray et al., 2024, Zhang et al., 2025b,a].

**Test-Time Scaling for Reasoning** A growing line of work boosts large-model reasoning by allocating extra compute *after* training. Early analyses show that properly budgeted TTS consistently raises accuracy across tasks [Snell et al., 2024]. Concretely, three main strategies have emerged. (i) *Best-of-n* reranks multiple sampled chains of thought (CoTs), as in BoNBoN and related variants [Gui et al., 2024]. (ii) *Guided decoding* steers beam search with learned value functions: outcome-supervised value models (OVM) [Yu et al., 2023] and self-evaluation-guided beams [Xie et al., 2023], are prominent examples. (iii) *Tree search* uses MCTS-style roll-outs—e.g. AlphaMath [Chen et al., 2024b]—to explore expansive reasoning spaces. Parallel efforts focus on verifier signals: Calibrated-CLIP augments vision–language models with paired positives/negatives [Zhou et al., 2024b], while LLaVA-Critic learns an open-source multimodal grader [Xiong et al., 2024]. Our approach departs from these text-centric methods by adding a *physically consistent world model*. The model renders imagined egocentric views, supplying geometry-aware evidence that guides search through a latent 3D scene, which help us perform TTS in spatial-reasoning tasks.

## 3 Approach

Our method targets test-time enhancement of vision–language models (VLMs) in 3D spatial reasoning by exploiting the predictive power of a world model. Our framework, MindJourney, achieves the test-time scaling through two tightly coupled components:

**Video-Diffusion Models as World Models.** Given a single RGB frame and an egocentric action sequence defined by camera pose, the world model synthesizes a coherent egocentric video that follows the given trajectory, effectively turning the still image into an explorable 3D world.

**Spatial Beam Search.** Guided by the spatial question, the VLM and the world model interactively search for helpful trajectories in the virtual 3D space defined by the given image.

Sec. 3.1 provides an overview of our test-time scaling pipeline. In Sec. 3.2, we define our formulation of the world model in MindJourney. In Sec. 3.3, we will introduce our iterative search algorithm, Spatial Beam Search, which uses a VLM and a world model to interactively explore the imagined 3D space. Finally, in Sec. 3.4, we introduce training and achitecture details of our world model, Search World Model (SWM), one of the world model candidates in our experiments.

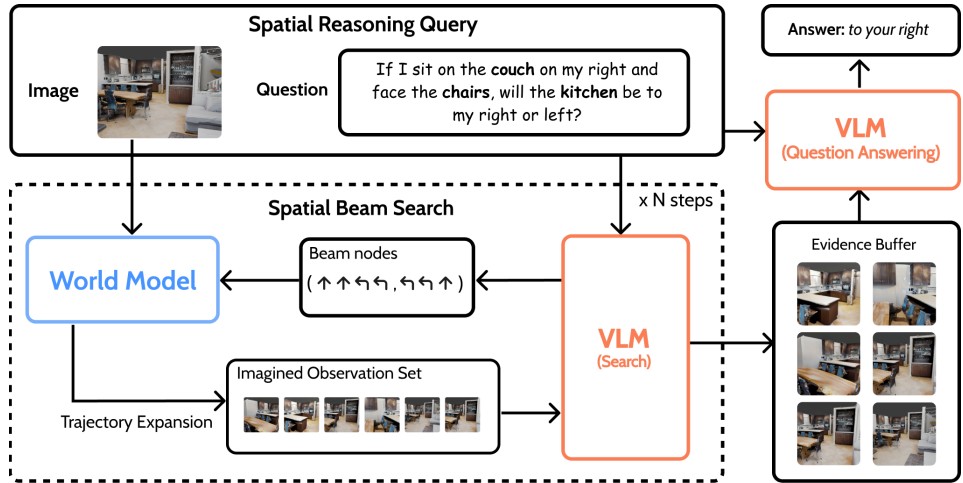

**Figure 2: MindJourney Pipeline.** Our pipeline starts with Spatial Beam Search for $n$ steps before answering the question. The world model interactively generates new observations, while a VLM constructs the evidence buffer for Q&A and guides the search during the process.

## 3.1 Pipeline Overview

Fig. 2 illustrates our pipeline: given a spatial reasoning query, the world model and the VLM collaborate through a Spatial Beam Search to generate and filter novel viewpoints that facilitate question answering. Up to three components participate: a World Model $\mathcal{W}$, a Search VLM $\mathcal{V}_{\text{search}}$, and a Question-Answering VLM $\mathcal{V}_{\text{QA}}$ (The VLMs may share the same network).

Given an input image and a spatial-reasoning query, we launch an $n$-step Spatial Beam Search instead of producing an answer directly. For every trajectory in the current beam, the world model expand each trajectories (Fig. 3), yielding candidate trajectories and their imagined observations. Conditioned on the query text, the search VLM evaluates the imagined observations and (i) writes trajectory–observation pairs that are highly relevant to the answer into a Helpful Observation Buffer, and (ii) selects trajectories worth further exploration to form the next beam layer.

After the search, the question-answering VLM consumes the original image together with the buffered observations to deliver the final answer to the spatial-reasoning query. This imagine → select → answer loop equips a frozen VLM with the world model's physical priors and motion forecasts, yielding substantial gains in 3D spatial reasoning without additional training.

## 3.2 World Model Formulation

We treat the world model as an egocentric simulator that rolls out a sequence of actions starting from a reference image.

**Action Space.** In our paper, we define the set of primitive actions to be

$$\mathcal{A} = \{\text{move-forward } d, \text{ turn-left } \theta_l, \text{ turn-right } \theta_r\}$$

, where $d$ is moving distance in meters and $\theta_l$ and $\theta_r$ are rotation angles in degrees. Further, we define a *action trajectory* as an ordered tuple

$$\boldsymbol{\tau} = (a_1, \ldots, a_m), a_i \in \mathcal{A}$$

, with length $m$. Therefore, we denote the search space of all trajectories of length at most $n$ by $\mathcal{T}_k = \bigcup_{m=0}^{n} \mathcal{A}^m$, where $\mathcal{A}^0 = \{\varnothing\}$ contains the empty trajectory.

**Action Representation.** Each primitive action $a \in \mathcal{A}$ is mapped to a relative camera-pose transformation $\varphi(a) = c \in \mathrm{SE}(3)$. Hence a trajectory $\boldsymbol{\tau} = (a_1, \ldots, a_m)$ is deterministically translated into the pose sequence

$$\boldsymbol{C} = (c_1, \ldots, c_m), \quad c_i = \varphi(a_i).$$

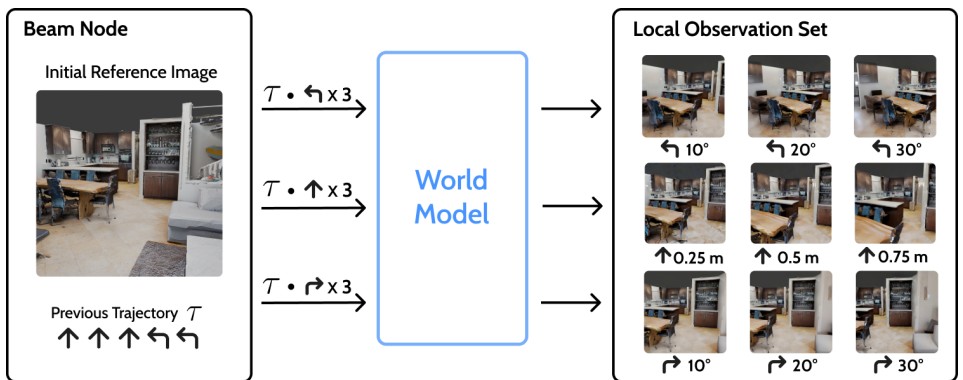

Figure 3: **Trajectory Expansion Illustration.** The Figure illustrate a Trajectory Expansion process with $k = 3$, $d = 0.25$, and $\theta = 10°$. In this case, the world model generates 9 new observations given the Beam Node.

We condition the video-diffusion world model on each $c_i$ to ensure that the $i$-th generated frame reflects the intended egocentric motion.

A pose $c$ is expressed by its intrinsic matrix $\mathbf{K}$ and extrinsic matrix $\mathbf{E} = [\,\mathbf{R} \mid \mathbf{t}\,]$.

**Action-Driven Video Generation.** Let $\mathbf{x}_0 \in \mathbb{R}^{H \times W \times 3}$ be the reference image and $\boldsymbol{C} = (c_1, \ldots, c_m) \in \mathrm{SE}(3)^m$ the pose sequence induced by a trajectory $\boldsymbol{\tau}$. Our pose–conditioned video diffusion model,

$$\mathcal{W} : \ (\mathbf{x}_0, \boldsymbol{C}) \ \longmapsto \ \mathbf{V} = (\mathbf{x}_1, \ldots, \mathbf{x}_m), \qquad \mathbf{x}_i \in \mathbb{R}^{H \times W \times 3},$$

maps the pair $(\mathbf{x}_0, \boldsymbol{C})$ to an *egocentric rollout* that follows the intended motion. As an outcome, the world model produces the synthetic video $\mathbf{V}$, as an imagined walk in the 3D space defined by the reference image.

### 3.3 Spatial Beam Search for Action Space Exploration

Our world model $\mathcal{W}$ can roll out an egocentric trajectory $\tau$—a sequence of primitive actions—in order to render the corresponding video frames. Because the discrete trajectory space $\mathcal{T}$ grows exponentially with its horizon, we employ a beam-search procedure that alternates between *question-agnostic expansion* and *question-aware pruning*. The search runs for at most $n$ steps; At each step we expand each current beam node and then invoke a vision–language model (VLM) to score the resulting candidates with respect to the spatial-reasoning question.

**Trajectory Expansion.** A beam node at depth $m$ stores a trajectory $\tau = (a_1, \ldots, a_m) \in \mathcal{T}_m$ ($\tau = \varnothing$ for the root). To search the action space starting from the beam node, we adopt the following strategy. For each primitive action $a' \in \mathcal{A}$ we permit up to $k$ consecutive repetitions, denoted $a'^{\,r}$ ($1 \le r \le k$). The candidate set generated from $\tau$ is

$$\mathcal{C}(\tau) \ = \ \big\{ \tau \oplus a'^{\,r} \mid a' \in \mathcal{A}, \ 1 \le r \le k \big\}, \qquad |\mathcal{C}(\tau)| \le 3k,$$

where $\oplus$ denotes concatenation. Each $\tau'^{\,3}$ is fed to the world model $\mathcal{W}$, yielding the next egocentric frame $\mathbf{x}_{\tau'} = \mathcal{W}(\mathbf{x}_0, \tau')$. We collect the resulting local observation set

$$\mathcal{O}(\tau) \ = \ \big\{ (\tau', \mathbf{x}_{\tau'}) \mid \tau' \in \mathcal{C}(\tau) \big\}.$$

The global observation set for the current search step is the union of $\mathcal{O}(\tau)$ over all beam nodes. In practice, because the world model $\mathcal{W}$ renders an entire video clip in one pass, we roll out only the candidates in $\mathcal{C}(\tau)$ whose length is exactly $m + k$—that is, trajectories that extend the current node by the full $k$ actions. A single call to $\mathcal{W}$ with this batched set then produces the complete local observation set $\mathcal{O}(\tau)$ for the node. Fig. 3 illustrate this process with an example.

---

[3]To avoid wasted rollouts we discard any candidate that (i) immediately reverses the previous action (e.g. turn left and then turn right) or (ii) exceeds a pre-set translation/rotation budget.

**Algorithm 1** Spatial Beam Search for Action Space Exploration

---

**Require:** Initial frame $\mathbf{x}_0$, world model $\mathcal{W}$, VLM $\mathcal{V}_{\text{search}}$, VLM $\mathcal{V}_{\text{QA}}$, primitive actions $\mathcal{A}$, spatial question $q$, parameters $n, k, B, H, \gamma_{\text{exp}}, \gamma_{\text{help}}$
**Ensure:** Final answer to question $q$
  1: Initialize beam $\mathcal{B} \leftarrow \{\varnothing\}$, evidence set $\mathcal{E} \leftarrow \emptyset$
  2: **for** $m = 1$ to $n$ **do**
  3:     $\mathcal{O} \leftarrow \emptyset$
  4:     **for all** $\tau \in \mathcal{B}$ **do**
  5:         Generate and prune candidates $\mathcal{C}(\tau)$ of length $|\tau| + k$
  6:         $\mathcal{O} \leftarrow \mathcal{O} \cup \{(\tau', \mathcal{W}(\mathbf{x}_0, \tau')) \mid \tau' \in \mathcal{C}(\tau)\}$
  7:     For each $(\tau', \mathbf{x}_{\tau'}) \in \mathcal{O}$, generate $\text{desc}(\tau')$
  8:     Query $\mathcal{V}_{\text{search}}$ for $s_{\text{exp}}(\tau'), s_{\text{help}}(\tau')$ using $\langle q, \text{desc}(\tau'), \mathbf{x}_{\tau'} \rangle$
  9:     Prune candidates below $\gamma_{\text{exp}}, \gamma_{\text{help}}$
10:     Update beam $\mathcal{B} \leftarrow \text{top-}B$ by $s_{\text{exp}}$
11:     Add top-$H$ by $s_{\text{help}}$ to evidence $\mathcal{E}$
12:     **if** $\mathcal{B} = \emptyset$ **then**
13:         **break**
14: Prepare evidence set $\{(\tau^{(h)}, \mathbf{x}_{\tau^{(h)}}, \text{desc}(\tau^{(h)}))\}_{h=1}^{H^*}$ from $\mathcal{E}$
15: Return final answer from $\mathcal{V}_{\text{QA}}$ given $q$ and the full evidence

---

**VLM-Based Heuristics.** At each search step, For every pair $(\tau', \mathbf{x}_{\tau'})$ in the global observation set, we create a natural-language description $\text{desc}(\tau')$ (e.g., "move forward 0.2 m, then turn right 30°") and feed the spatial reasoning question and all the tuples $\langle q, \text{desc}(\tau'), \mathbf{x}_{\tau'} \rangle$ from the observation set to the VLM. The model returns two scalar logits for each $\tau'$ through two different criteria:

$$s_{\text{exp}}(\tau') \quad \text{("how useful is it to \textit{keep exploring} this trajectory?"),}$$

$$s_{\text{help}}(\tau') \quad \text{("how useful is \textit{this view} for answering now?").}$$

We first discard any pair whose score falls below fixed thresholds $\gamma_{\text{exp}}$ or $\gamma_{\text{help}}$, respectively. Among the remaining candidates we (i) retain the top $B$ by $s_{\text{exp}}$ as the next-step beam (beam width $B$), and (ii) cache the top $H$ by $s_{\text{help}}$ as "helpful" viewpoints and save them to the evidence buffer for the final answer. We thus use VLM to drive question-aware pruning of the search tree and accumulates evidence views that will later be supplied to the VLM for answer generation.

**Answer Generation.** After the search step limit $n$ or running out of beam node due to thresholding, we collect all cached helpful trajectories $\{\tau^{(h)}\}_{h=1}^{H^*}$ together with their associated frames and natural-language descriptions. The VLM receives the original question and this multi-view evidence in a single pass and outputs the final answer. We present our algorithm in Algorithm 1.

## 3.4 Search World Model Details

We trained our own world models, Search World Model (SWM), specifically for the defined action space in Sec. 3.2. Please refer to the Appendix for more details.

**Architecture.** SWM is based on the Wan2.2-TI2V-5B [Wan et al., 2025], following ReCamMaster [Bai et al., 2025]. Specifically, we represent the camera transform by camera extrinsic matrices and directly add the embeded camera matrices to the video latent in a pixel-wise manner.

**Training Dataset.** The world model only has to execute the limited set of primitive egocentric actions used by MindJourney, so we synthesise most of its training corpus with the Habitat 2.0 navigation simulator [Szot et al., 2022]. Habitat provides pixel-accurate renderings for forward, backward and rotational motions in indoor environments, making it ideal for learning precise camera control. To bridge the appearance gap between synthetic interiors and real imagery, we blend the Habitat clips with two large-scale, view-consistent video datasets—RealEstate-10K and DL3DV-10K [Ling et al., 2024]. The resulting mix couples Habitat's geometric fidelity with the visual diversity of real indoor and outdoor scenes, allowing the world model to generalise beyond its synthetic training domain.

Table 1: **SAT-Real.** Accuracy for large proprietary and open-source MLMs on SAT-Real. Specifically, OpenAI o1 has test-time scaling capability. MJ refers to MindJourney, augmented on both SWM and SVC. Results marked with * are from [Ray et al., 2025].

| | | SAT Real | | | | |
|---|---|---|---|---|---|---|
| | Avg | EgoM | ObjM | EgoAct | GoalAim | Pers |
| GPT4-V* | 50.7 | - | - | - | - | - |
| Gemini1.5-flash* | 57.6 | - | - | - | - | - |
| Gemini1.5-pro* | 64.8 | - | - | - | - | - |
| Robopoint-13B* | 46.6 | - | - | - | - | - |
| GPT-4o | 60.3 | 56.5 | **85.0** | 50.0 | 64.0 | 45.0 |
| + MJ (SWM) | 70.6 | 60.9 | 56.5 | 75.7 | 85.3 | 66.7 |
| + MJ (SVC) | 69.3 | 78.3 | 60.9 | 78.4 | 70.6 | 57.6 |
| GPT-4.1 | 74.0 | 95.7 | 73.9 | 78.3 | 88.2 | 39.4 |
| + MJ (SWM) | 80.6 | **100.0** | 78.3 | 89.2 | **91.2** | 48.4 |
| InternVL3-14B | 59.3 | 56.5 | 69.5 | 54.0 | 73.5 | 45.4 |
| + MJ (SWM) | 66.6 | 82.6 | 60.9 | 67.5 | 82.4 | 42.4 |
| OpenAI o1 | 74.6 | 78.3 | 82.6 | 73.0 | 73.5 | 69.7 |
| + MJ (SWM) | **84.7** | 95.7 | 82.6 | 83.8 | 88.2 | **75.8** |
| + MJ (SVC) | 77.3 | **100.0** | 65.2 | 78.4 | 82.4 | 63.7 |

# 4 Experiment

## 4.1 Experiment Settings

**Benchmarks.** Our main benchmark is the Spatial Aptitude Training (SAT) benchmark, which probes an agent's ability to reason about both egocentric motion and object motion—key skills for embodied AI. SAT is split into SAT-Synthesized, 4000 synthetic questions rendered in AI2-THOR [Kolve et al., 2017] indoor scenes, and SAT-Real, real images spanning indoor and outdoor environments. The two splits therefore cover a wide distribution shift, letting us evaluate both in-distribution accuracy and real-world transfer.

**Evaluation Metrics.** As the SAT benchmark are all multiple choices questions, we use accuracy as our evaluation metric accross all tasks.

**Vision-Language Models.** We pair our pipeline with four representative VLMs: the closed-source GPT-4o and GPT-4.1 (strong general-purpose multimodal baselines), InternVL3-14B (one of the most capable open-source VLMs to date), and o1, a reinforcement-learning–fine-tuned model that performs test-time chain-of-thought scaling. This mix spans both proprietary and fully open ecosystems as well as models with and without explicit reasoning scaffolds.

**World Models.** Our experiments use two distinct video world models. (i) Search World Model (SWM), the world model we trained, introduced in Sec. 3.4; and (ii) Stable-Virtual-Camera (SVC), a recently released, publicly available generator that produces geometrically stable novel views.

**Spatial Beam Search Configurations.** Unless noted otherwise, we use the same search configuration for every experiment: search depth $n = 3$ steps; up to $k = 3$ consecutive repetitions per primitive action during each expansion; exploration and helpfulness thresholds $\gamma_{exp} = 8$, $\gamma_{help} = 8$.

## 4.2 Experiments Results

The SAT benchmark comprises five spatial-reasoning tasks—ego movement (EgoM), object movement (ObjM), action consequence (EgoA), and perspective shifts (Pers)—each mirroring challenges an embodied agent routinely faces in 3D space. We evaluate our method on both SAT-Real and SAT-Synthesized and observe a clear performance boost over all baselines on both splits.

**Baselines.** Our baselines are the four VLMs: GPT-4o, GPT-4.1, InternVL3-14B, and o1. Despite their diversity—closed source vs. open source, standard training vs. RL-based test-time scaling—we expect each to benefit from our approach, MindJourney. For the o1 experiments, for saving computa-

Table 2: **SAT-Synthesized.** Accuracy for large proprietary and open-source MLMs on SAT-Synthesized. Specifically, OpenAI o1 has test-time scaling capability. MJ refers to MindJourney, augmented on both SWM and SVC.

| | **SAT Synthesized** | | | | | |
| | **Avg** | **EgoM** | **ObjM** | **EgoAct** | **GoalAim** | **Pers** |
|---|---|---|---|---|---|---|
| GPT-4o | 61.0 | 64.7 | 86.8 | 51.9 | 68.7 | 43.4 |
| + MJ (SWM) | 70.8 | 77.6 | 82.6 | 70.1 | 84.5 | 45.8 |
| + MJ (SVC) | 72.3 | 80.0 | 84.8 | 65.0 | **89.3** | 51.4 |
| GPT-4.1 | 66.4 | 75.3 | 89.0 | 57.8 | 78.3 | 41.5 |
| + MJ (SWM) | 75.4 | **88.2** | **92.4** | 70.8 | **89.3** | 45.8 |
| InternVL3-14B | 61.4 | 74.6 | 85.9 | 53.3 | 84.5 | 20.6 |
| + MJ (SWM) | 66.6 | 69.4 | 58.2 | 68.1 | 81.9 | 58.0 |
| OpenAI o1 | 72.4 | 78.0 | 85.9 | 65.4 | 86.0 | 54.6 |
| + MJ (SWM) | 76.8 | 82.4 | 80.4 | **76.6** | 88.1 | 60.7 |
| + MJ (SVC) | **78.6** | 87.1 | 80.4 | 72.3 | **89.3** | **70.1** |

tional cost, we use o1 only as the question-answering VLM and let GPT-4o handle the search phase. In every other experiment the same VLM is used for both search and the final Q & A.

**SAT-Real.** The SAT-Real split comprises 150 real-image queries spanning indoor and outdoor scenes. Table 1 shows that augmenting each VLM with MindJourney results in uniform and sizeable gains. Average top-1 accuracy rises by 7.7 percentage points, and GPT-4o achieves the largest single boost at over 10%. Remarkably, GPT-4.1 with our method already surpasses vanilla o1; when *o1* itself is paired with MindJourney it sets a new state of the art on SAT-Real. These findings confirm that world-model-driven test-time scaling complements RL-based scaling and our method generalizes to real, outdoor scenarios.

**SAT-Synthesized.** Because the synthetic split contains 4 000 questions, we evaluate on a random 500-question subset to keep the *o1* runs tractable. Results in Table 2 reproduce the pattern seen on real images: mean accuracy improves by 8.0% in average. Again, the performance of GPT-4.1 with our method surpasses vanilla o1, and o1 with our method improves its performance. Moreover, across all five SAT question types the highest score is always achieved by a MindJourney-augmented model, underscoring that the proposed framework consistently improves reasoning across all question types.

### 4.3 Ablation Study

In our ablation study, we ablate the hyperparameters of our search method. Specifically, for gpt-4o augmented with SWM, we ablate with search step $n \in \{1, 2, 3\}$ and VLM pruning threshold $\gamma \in \{4, 6, 8\}$. We experimented on these values on both SAT-Real and SAT-Synthesized. Accuracy is reported on both SAT-Real and the 500-question SAT-Synthesized subset, allowing us to evaluate how each setting affects performance in real-image and synthetic scenarios.

**Search Depth.** Panel (a) of Fig. 4 shows that accuracy for threshold 4 and 6 on SAT-Real peaks at search steps 2 and then drops slightly at step 3. The decline stems from the limits of our world model, SWM: trained predominantly on indoor synthetic data, it can struggle to faithfully simulate outdoor views once the imagined trajectory strays too far from the reference frame. On SAT-Synthesized—whose scenes are closer to the SWM's training distribution—performance continues to rise through step 3 for both threshold 8 and 6, confirming that deeper exploration remains beneficial when the world model can faithfully render long roll-outs. More visual results can be found in the Appendix.

**Pruning Threshold.** Panels (a) and (b) also show the importance of the VLM score threshold. A lenient threshold lets many low-quality views into the evidence buffer, diluting the signal and lowering accuracy. The effect is stronger on the Real split, where generation quality is lower, so the threshold has a larger impact there.

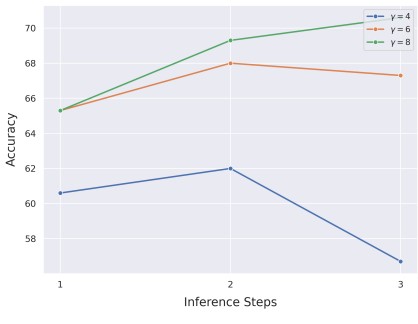

(a) Accuracy on SAT-Real.

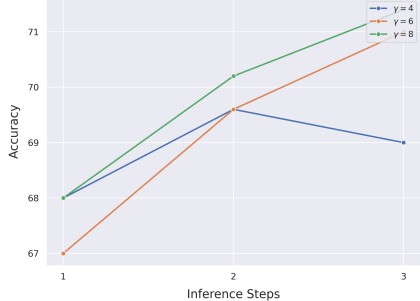

(b) Accuracy on SAT-Synthesized.

Figure 4: **Inference Steps vs. Accuracy.** Accuracy on SAT-Real and SAT-Synthesized with different VLM Thresholds and inference steps.

## 4.4 Analysis

**Test-time Scaling with World Models vs. Test-time Scaling with RL.** On both SAT splits, a plain VLM augmented with our world-model search already surpasses the RL-fine-tuned o1. When the same world-model search is applied on top of o1, accuracy climbs still higher, yielding the best results overall. These two observations imply that the exploratory roll-outs supplied by the world model provide information that is largely orthogonal to the inductive bias learned through RL chain-of-thought. The phenomenon highlights an exciting potential: giving a reasoning engine a physically consistent imaginary workspace can enhance, rather than replace, other forms of test-time self-improvement.

**World Model Capability.** The ablation results reveal a clear bottleneck: once the imagined trajectory strays too far from the initial frame, today's world models begin to break down in terms of generation quality. Fig. 4a shows that this degradation not only lowers the fidelity of rendered views but also feeds noisy evidence to the VLM, ultimately capping the benefits of deeper search. Although our world model SWM and the state-of-the-art Stable-Virtual-Camera (SVC) perform similarly within the three-step regime explored here, further gains will require world models that can sustain geometric and photometric consistency over much longer roll-outs.

## 5 Conclusion

We have presented MindJourney, the first framework that equips a vision–language model with a world model for imagination at test-time Through Spatial Beam Search, the VLM actively explores the latent 3D scene behind a single image, caching the most informative imagined views for spatial reasoning. This simple, training-free procedure improves four heterogeneous VLMs—ranging from closed-source GPT-4o to the RL-scaled *o1*—to new state-of-the-art accuracy on the SAT benchmark. Gains are robust across both synthetic and real images, across all SAT task categories, and across two different world-model generators, underscoring the model-agnostic nature of our approach.

Beyond its empirical performance, MindJourney offers a conceptual advance: it shows that for spatial reasoning, giving a reasoning engine a physically consistent simulator at test time can complement, and even surpass, complex RL-based self-reflection pipelines.

**Limitations and Future Works** Our current pipeline assumes a single reference view. When a spatial-reasoning query supplies multiple images, MindJourney fails to treat the extra views as entry points into the scene. An ideal system would regard each image as a separate "portal," launch an exploration from every portal, and fuse the resulting evidence. Extending our Spatial Beam Search to a multi-source setting is therefore a natural next step.

A second limitation lies in the *question-agnostic* nature of today's controllable video world models. Because the generator is unaware of the downstream query, it can hallucinate views that are irrelevant—or even contradictory—to what the question implicitly assumes. Future work should develop query-conditioned world models or incorporate lightweight constraint mechanisms so that imagined roll-outs remain consistent with the task at hand.

## Acknowledgments and Disclosure of Funding

We are grateful to Anushka Agarwal for assistance with the baseline code, and to Jiaben Chen, Zeyuan Yang, Lixing Fang, Haoyu Zhen, and many other friends for their helpful feedback and insightful discussions.

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

# A   Experiment Details

## A.1   Inference Details

### A.1.1   Search Configurations

We use the same search configuration for every experiment: a search depth of $n = 3$ steps, a beam size of 2, exploration and helpfulness thresholds $\gamma_{\text{exp}} = 8$ and $\gamma_{\text{help}} = 8$, and a maximum trajectory length of 8 starting from the given reference image. During each expansion, we allow up to $k = 3$ consecutive repetitions per primitive action; each forward step moves the agent 0.25m and each rotation step turns it by $9°$. We prepared more visual examples about the Trajectory Expansion process under our experiment settings at Fig. 5 and Fig. 6

### A.1.2   Computational Resources

All inference experiments were run on high-performance NVIDIA GPUs: when using Search World Model as the world model, we employed A40 GPUs with 40GB of VRAM; when using Stable-Virtual-Camera as the world model, we ran on H100 GPUs with 80GB of VRAM; and for all experiments combining the InternVL3-14B VLM with the Search World Model, we also used H100 GPUs to accommodate the larger memory footprint of the vision–language model.

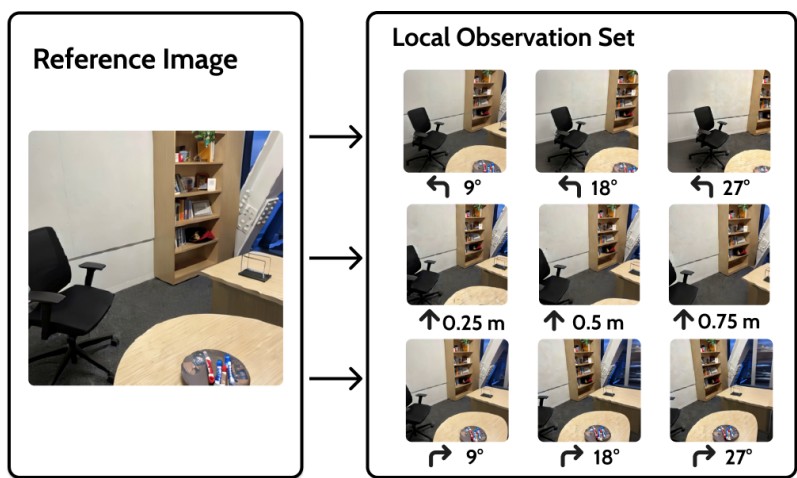

Figure 5: Trajectory Expansion example on SAT-Real.

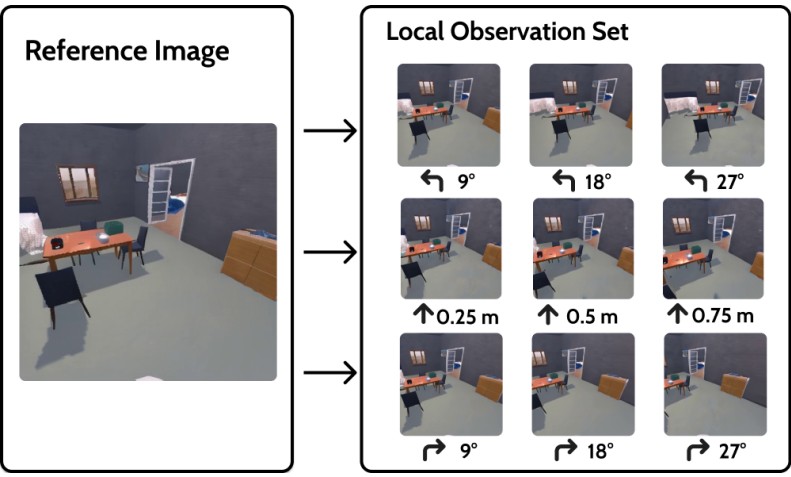

Figure 6: Trajectory Expansion example on SAT-Synthesized.

## A.2 Search World Model Training

### A.2.1 Dataset

The training set for our Search World Model (SWM) comprises three components: HM3D, DL3DV-10K, and RealEstate10K [Szot et al., 2022, Ling et al., 2024, Zhou et al., 2018].

**HM3D.** We generate 50K simulated navigation clips using Habitat on HM3D scenes. For each episode, we sample a random start and goal position and follow the shortest path for up to 500 steps. To improve robustness to camera tilt, we uniformly draw an initial pitch from $[-30°, 20°]$, and in 10% of episodes we hold the agent at a fixed location and vary only its pitch by rotating up and down.

**RealEstate10K.** This collection of 10K real indoor videos enriches our data with diverse residential environments, balancing the simulated HM3D distribution.

**DL3DV-10K.** Similarly, we incorporate 10K real outdoor videos from DL3DV-10K to capture natural scenes and broaden our model's generalization to exterior settings.

To ensure consistent camera dynamics across all sources, we normalize each clip's frame rate and spatial resolution before training.

### A.2.2 Implementation Details

We adopt Wan2.2-TI2V-5B [Wan et al., 2025] as our backbone. We followed the implementation of ReCamMaster, which conditions video generation on the target camera poses by encoding each frame's extrinsic matrix (3×4 rotation–translation) with a small learnable camera encoder and adding this signal to the visual features inside every diffusion-transformer block, right after spatial attention and before 3D (spatio-temporal) attention. During the training, we only finetuned the camera encoder and a newly introduced projector. For more details, please refer to ReCamMaster's open-sourced codebase.

### A.2.3 Training Details

We subsample training clips with a frame-skip stride uniformly drawn between 1 and 3 to expose the model to varied camera motions. Optimization is performed with Adam and a linear warmup schedule to a peak learning rate of $3e-5$, using bfloat16 precision for efficiency and clipping gradients to a maximum norm of 1.0 for stability. All video-diffusion models are trained on eight NVIDIA H100 GPUs over approximately three days.

## B Failure Case Analysis

Despite the strong overall performance of our model, we conducted a detailed analysis of the failure cases to uncover its limitations. The following examples highlight typical scenarios where the model does not perform as expected.

Unlike the previous figures, the action labels in Fig. 7 represent delta action at each step.

### B.1 World Model Capabilities

#### B.1.1 Case 1: Inaccurate Forward Movement

In group (a), the imagined trajectories systematically under- or over-estimate forward translations: the actual step lengths no longer match the intended distances, and the displacement between successive frames varies unpredictably. As a result, the agent repeatedly overshoots its targets and exhibits jittery, erratic motion within the simulated environment.

We hypothesize that these errors arise from scale inconsistencies across our training sources. In the Search World Model, only HM3D provides metrically accurate movement distances, whereas the Stable Virtual Camera relies on datasets with more heterogeneous scale calibrations. When these conflicting scale conventions are combined during training, the model learns incompatible motion priors—manifesting exactly as the misaligned, erratic forward movements seen in group (a).

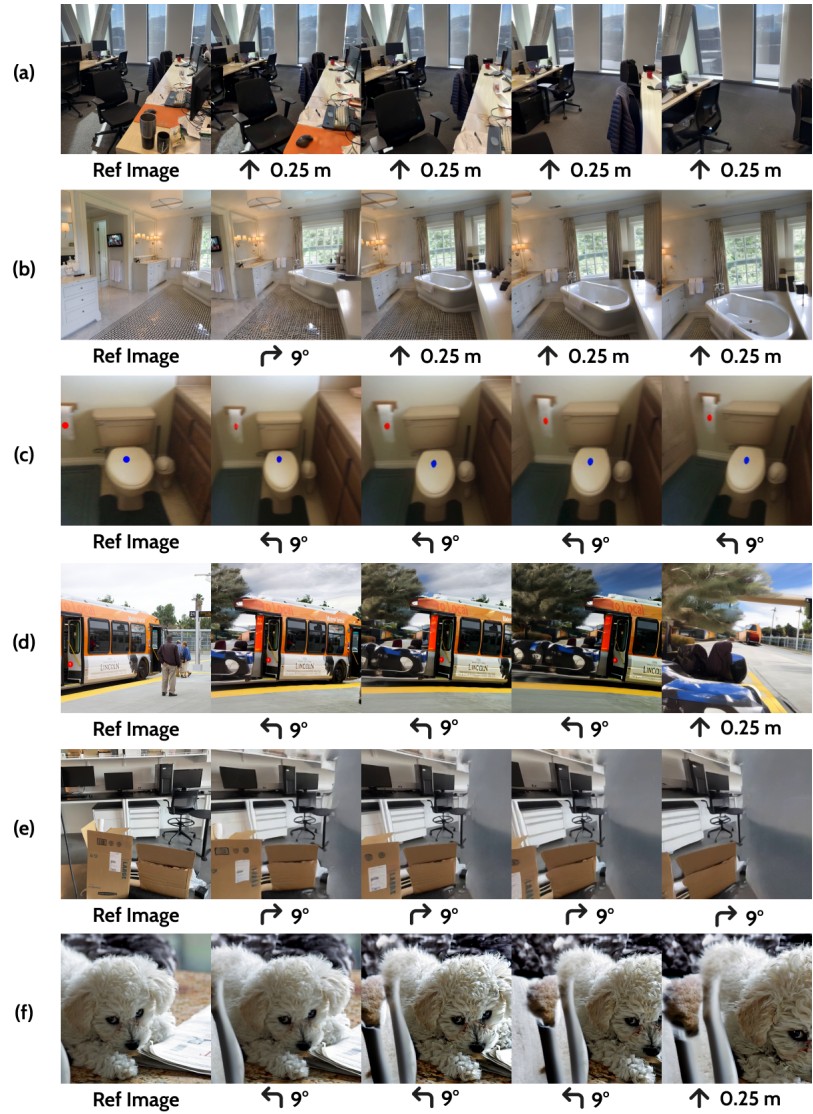

Figure 7: **Failure Cases of World Models**. Group (a) shows inaccueate forward movement; group (b) shows unintended roll movement leading to a tilted scene; group (c) shows the unstable egocentric rotation that introduces viewpoint movement; group (d) shows the model generate artifacts for unseen regions; group (e) shows misinterpretation when inference on real-world scene; group (f) shows a failure case on out-of-domain animal data.

### B.1.2 Case 2: Unintended Roll Movement

In group (b), the predicted images exhibit an unnatural tilt of the scene, where the horizon line is significantly misaligned. This indicates that the model sometimes introduces unintended roll movements, resulting in a distorted camera orientation.

### B.1.3 Case 3: Unstable Egocentric Rotation

In group (c), the predicted images exhibit unstable viewpoints during egocentric rotation. The transitions between consecutive frames are inconsistent, and the visual perspective appears to undergo a rightward translation while simultaneously rotating.

This issue happens more often for our world model SWM, which happens because we blend some RealEstate10K data when fine-tuning SWM. RealEstate10K contains numerous segments in which the camera trajectory exhibits simultaneous translation motion and rotation, leading to a distributional bias in training and causing the model to develop systematic prediction errors.

### B.1.4 Case 4: Visual Artifacts

In group (d), the predicted images contain noticeable visual artifacts, particularly in regions that are occluded or unseen in the input view. These artifacts manifest as texture distortions, unnatural edges, or inconsistent object boundaries, which significantly degrade the visual realism of the generated images.

This issue may stem from the model's limited ability to hallucinate plausible content in areas with insufficient visual context or out of domain data. In particular, when the target view includes regions not visible in the source image, the model may rely on weak priors or overfit to spurious patterns seen during training.

### B.1.5 Case 5: Out of domain data - scene misinterpretation

In group (e), the model exhibits clear failures when processing scenes that fall outside the distribution of the training data. The predicted images demonstrate significant misinterpretation of scene structure, such as incorrect boundary extension as shown in the example. These failures are especially prominent in complex real-world environments with lighting, textures, or layouts not observed during training.

We attribute this behavior to the model's limited generalization ability when confronted with out-of-distribution inputs. Without adequate exposure to diverse scene types during training, the model tends to rely on learned priors that do not transfer well, resulting in hallucinated or semantically inconsistent content.

### B.1.6 Case 6: Out of domain data - human or animal

In group (f), as shown in the images, the body of the dog is missing. The model fails to generate plausible predictions when encountering humans or animals, which are underrepresented or absent in the training data. The generated images often exhibit severe distortions in body shape or texture consistency, making the predictions semantically incorrect.

This failure can be attributed to the model's lack of exposure to articulated and deformable entities during training. Humans and animals involve complex structures and dynamic poses that require specialized representation and learning. Without sufficient domain-specific data, the model struggles to generalize, leading to implausible reconstructions or complete semantic failures.

## B.2 VLM Capabilities

### B.2.1 Case 1: VLM Q&A

As illustrated in Fig. 8, although the world model generates great visualizations that would intuitively help the VLM with spatial reasoning, the VLM can still be confused and cannot answer the question correctly. Therefore, for spatial reasoning question, the question answering ability of the base VLM is still very important.

**Question:**
If I turn left by 16 degrees, will I be facing away from GarbageCan (near the mark 3 in the image)?
**Answer Choices:**
['yes', 'no']

Correct Answer: yes
LLM Response: no (wrong)

These are the images that pair with the question.
Image 1:

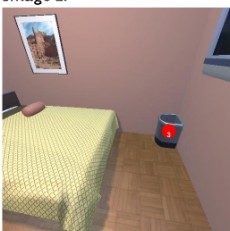

Image 1 is your current egocentric view

Below are the imagined views you would obtain if you took the corresponding actions. These are provided to help you answer the question.

Action: turn left
turn left 18 degrees

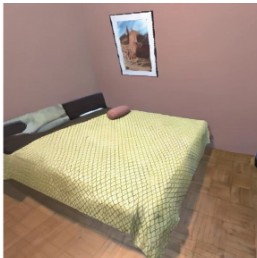

turn left 27 degrees

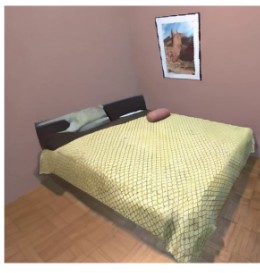

Figure 8: Failure case - VLM's Q&A ability is not sufficient.

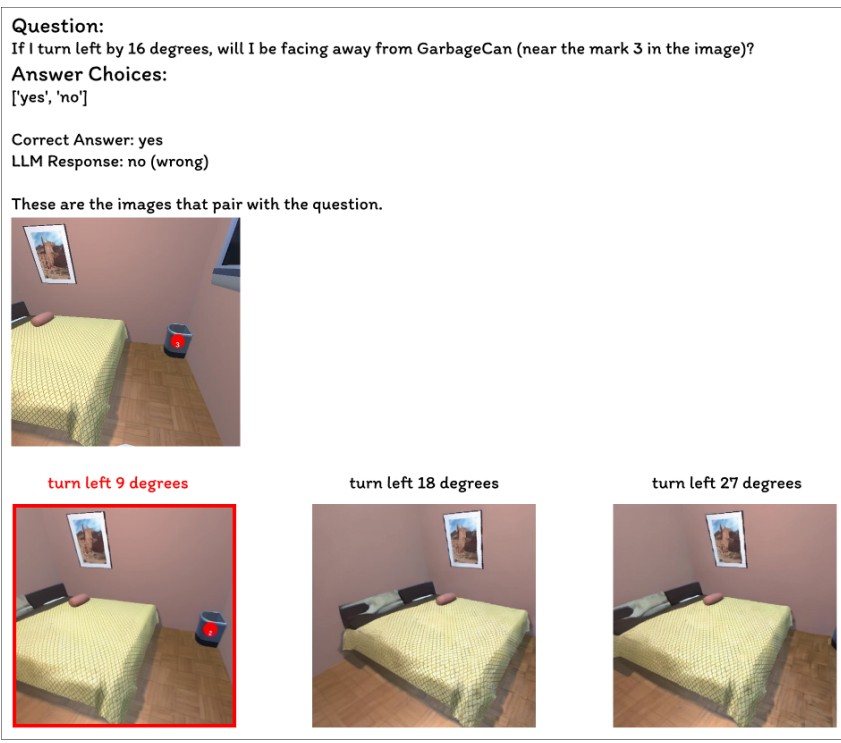

Figure 9: Failure case - VLM's scoring ability is not sufficient.

### B.2.2 Case 2: VLM Scoring

Given the same question mentioned in Case 1, the VLM is not able to keep one of the most informative image after the scoring process. As illustrated in Fig. 9, the "turn left 9 degrees" is the most informative image as it contains the garbage can mentioned in the question. However, the VLM scoring process does not keep the image in the final evidence buffer, which leads to a wrong answer. The improvement of VLM capability will also benefit the VLM scoring process and improve the overall performance implicitly.

## C  More Ablation Studies

### C.1  World Models

We evaluated the performance of two world models, Search World Model (SWM) and Stable Virtual Camera (SVC), on a dataset generated through the AI2-THOR simulator, as AI2-THOR is out-of-domain for both world models. The evaluation includes both quantitative metrics, measuring the accuracy of predictions and the quality of the generated images, and qualitative comparisons through visualizations of representative samples.

During inference, both models are executed with 50 diffusion steps. Specifically, to get the metrics of generated quality, we generated 10 episodes for each of the 208 scenes in AI2-THOR. Each episode consists of an action sequence of 8 steps, where at each step, an action is randomly selected from the primitive action set:
{move forward 0.25 meter, turn left 9 degrees, turn right 9 degrees}

### C.1.1  Quantitative Comparison

More specifically, following the previous work of stable virtual camera, we tested the prediction result of our world models using standard metrics-peak signal-to-noise ratio (PSNR), learned perceptual image patch similarity (LPIPS), and structural similarity index measure (SSIM). Results are shown in

Table 3: Video Generation Results. Comparison of SWM and SVC in both visual quality and consistency.

| Method | PSNR ↑ | SSIM ↑ | LPIPS(1e-4) ↓ |
|---|---|---|---|
| SVC | 64.51±0.27 | 0.994±0.01 | 0.49±0.01 |
| SWM | 66.59±0.21 | 0.997±0.01 | 0.31±0.01 |

Table 3, a quantitative comparison between two video generation models, SWM and SVC. These metrics jointly assess both visual fidelity and perceptual consistency.

SWM outperforms SVC in terms of PSNR (66.59 vs. 64.51) and SSIM (0.997 vs. 0.994), indicating more accurate and structurally consistent predictions. It also achieves a lower LPIPS score (0.31 vs. 0.49), suggesting that SWM generates images that are more perceptually similar to the ground truth. Overall, SWM demonstrates superior performance in terms of visual accuracy and perceptual similarity. This suggests that SWM is more effective at generating coherent and visually faithful video sequences for the primitive actions we defined.

### C.1.2 Qualitative Comparison

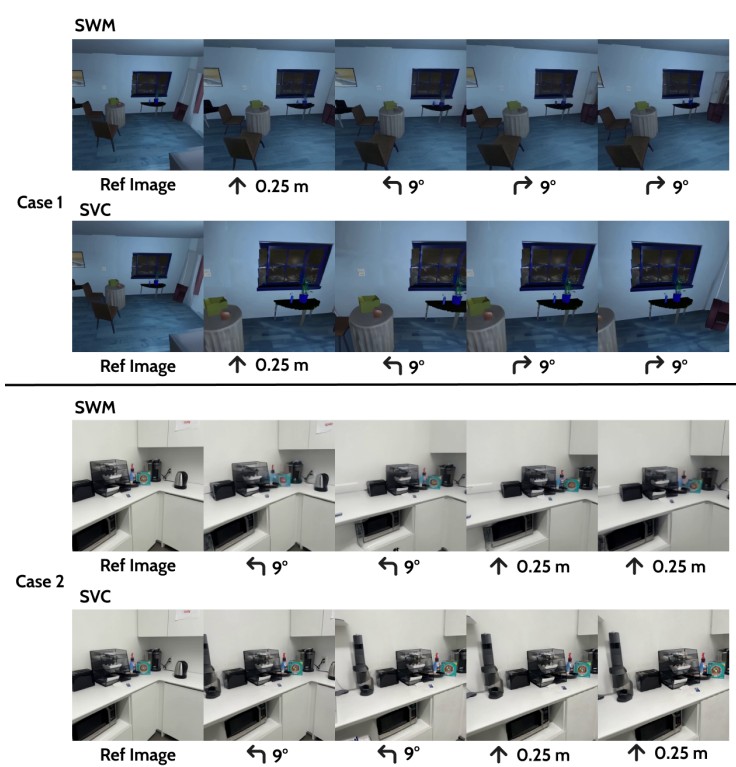

Figure 10: **Comparison of World Models**. Case 1 comes from validation split of SAT dataset; Case 2 comes from real-world test split of SAT dataset.

We present a qualitative comparison between the two models, SWM and SVC, using two representative examples from the synthesized validation set and real-world test set of SAT dataset. As shown in Case 1 from Fig.10, SVC sometimes performs inaccurate forward motion. After moving forward by 0.25m, while the visual consequence from SWM seems reasonble, the outcome from SVC shows its inconsistency. In Case 2 Fig.10, the SVC shows better capability of keeping object-level details. In the generated results from SWM, the objects become blurry as the video extends, but SVC successfully keeps details of each existing object. Generally, we observed that while SWM is more consistent in the scale of movement, SVC preserves more details during the camera movements.

## C.2  Ablation on Trajectoy Description

In our current method, for each observation in the evidence buffer, we also provide a natural language trajectory description that explains its relationship with the initial reference image. We demonstrate that the trajectory description is necessary for our method in Table 5 and Table 6. According to the tables, we observe that the performance of our method drops on both SAT-Real and SAT-Synthesized for all VLMs and world models.

## C.3  Ablation on Time Consumption

To assess the time efficiency of our method, we evaluated it with GPT-4o as the VLM while varying the number of inference steps. We also reduced the SWM's video-generation iterations from 50 to 20 to examine the trade-off between runtime and accuracy. The results show that fewer SWM iterations markedly speed up inference but degrade the quality of the generated video, which in turn lowers accuracy. Conversely, increasing the number of inference steps forces the SWM to predict more frames and provides more images to the VLM, thereby increasing both the SWM time and the VLM search/QA times.

However, we also observe that our method does not degrade significantly with 2 search steps and 20 iterations. Therefore, while our current setting optimizes for performance, step 2 and iter 20 is a much more cost-effective and recommended setting practically.

Table 4: SWM/VLM Time Consumption and Accuracy

| Setting | SWM time(s) | Search time(s) | Q&A time(s) | Total(s) | ACC(%) |
|---|---|---|---|---|---|
| step:1 iter:20 | 11.49 | 3.04 | 1.74 | 16.27 | 64.7 |
| step:2 iter:20 | 19.68 | 5.53 | 1.71 | 26.93 | 68.6 |
| step:3 iter:20 | 29.37 | 8.00 | 3.24 | 40.60 | 69.3 |
| step:1 iter:50 | 30.57 | 2.94 | 1.84 | 35.36 | 65.3 |
| step:2 iter:50 | 63.62 | 5.52 | 1.79 | 70.93 | 69.3 |
| step:3 iter:50 | 149.75 | 8.30 | 3.43 | 161.48 | 70.6 |

# D  Prompts

Here we provide 4 different prompts used in our method. The baseline prompt is shown in Fig. 11. The exploration Scoring prompt is shown in Fig. 12. The helpful Scoring prompt is shown in Fig. 13. The question-answering prompt using MindJourney is shown in Fig. 14.

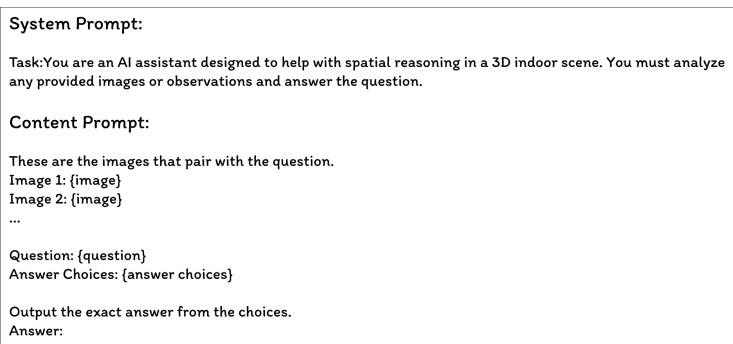

Figure 11: **Prompt for Baseline Q&A.**

Figure 12: **Prompt for Exploration Scoring.**

Figure 13: **Prompt for Helpful Scoring.**

Table 5: **Ablation on Trajectory Description.** Accuracy for large proprietary and MLMs on SAT-Real.

| | SAT Real | | | | | |
|---|---|---|---|---|---|---|
| | Avg | EgoM | ObjM | EgoAct | GoalAim | Pers |
| GPT-4o | 60.3 | 56.5 | 85.0 | 50.0 | 64.0 | 45.0 |
| + MJ (SWM) | 68.0 | 73.9 | 69.6 | 75.7 | 73.5 | 48.5 |
| + MJ (SWM) , w/o Traj. Desc. | 66.8 | 60.0 | 76.7 | 71.0 | 70.0 | 48.3 |
| + MJ (SVC) | 69.3 | 78.3 | 60.9 | 78.4 | 70.6 | 57.6 |
| + MJ (SVC) , w/o Traj. Desc. | 66.5 | 73.5 | 65.0 | 74.5 | 66.3 | 53.1 |
| GPT-4.1 | 67.3 | 81.0 | 76.4 | 69.5 | 73.9 | 36.0 |
| + MJ (SWM) | 82.6 | 95.0 | 78.2 | 89.0 | 85.0 | 66.6 |
| + MJ (SWM), w/o Traj. Desc. | 73.0 | 100.0 | 78.2 | 67.7 | 75.8 | 53.1 |

Table 6: **Ablation on Trajectory Description.** Accuracy for large proprietary MLMs on SAT-Synthesized.

| | SAT Synthesized | | | | | |
|---|---|---|---|---|---|---|
| | Avg | EgoM | ObjM | EgoAct | GoalAim | Pers |
| GPT-4o | 61.0 | 64.7 | 86.8 | 51.9 | 68.7 | 43.4 |
| + MJ (SWM) | 70.8 | 77.6 | 82.6 | 70.1 | 84.5 | 45.8 |
| + MJ (SWM) , w/o Traj. Desc. | 64.1 | 67.9 | 83.1 | 63.0 | 71.8 | 42.3 |
| + MJ (SVC) | 72.3 | 80.0 | 84.8 | 65.0 | 89.3 | 51.4 |
| + MJ (SVC) , w/o Traj. Desc. | 65.8 | 64.7 | 83.3 | 68.4 | 65.8 | 47.8 |
| GPT-4.1 | 66.4 | 75.3 | 89.0 | 57.8 | 78.3 | 41.5 |
| + MJ (SWM) | 75.4 | 88.2 | 92.4 | 70.8 | 89.3 | 45.8 |
| + MJ (SWM), w/o Traj. Desc. | 72.6 | 89.3 | 92.3 | 65.0 | 88.1 | 33.4 |

# E    Broader Impacts

By allowing vision–language models to build and interrogate a physically consistent "mental workspace," our method could accelerate progress in assistive robotics, remote inspection, and immersive training: robots that better understand 3D space can navigate cluttered homes for elder care, inspect hazardous sites without human entry, and deliver richer AR/VR experiences for education or therapy. At the same time, safer decision-making from imagined roll-outs may reduce real-world trial-and-error, lowering both cost and risk. Yet the technology also raises concerns. More capable spatial reasoning can enhance autonomous surveillance systems or military platforms; and greater autonomy could displace certain manual-labor jobs. Finally, training large video world models consumes considerable energy and inherits any biases present in the data (e.g., under-representation of certain environments). Researchers and practitioners should therefore pair technical advances with robust provenance tracking for generated content, scenario-specific safety constraints, and data-diversity audits, while favouring energy-efficient architectures and openly reporting compute footprints.

---

**System Prompt:**

Task: You are an AI assistant designed to help with spatial reasoning in a 3D indoor scene. You must analyze any provided images or observations and answer the question.
Rules:
   1. You should output the exact answer from the choices.
   2. You will be provided with multiple imagined views if you take corresponding actions to help you answer the questions.
   3. You can include minimal reasoning, but your final line must only include the exact answer choice.

**Content Prompt:**

These are the images that pair with the question.
Image 1: {image}
Image 2: {image}
...
Image 1 is your current egocentric view.

Question: {question}
Answer Choices: {answer choices}
Below are the imagined views you would obtain if you took the corresponding actions. These are provided to help you answer the question.
You can include them in your reasoning, but you should still only output the exact answer at the last line.

Action: {action catalog}
{action sequence}
{image}

Action: {action catalog}
...

Output the exact answer from the choices.
Answer:

---

Figure 14: **Prompt for Q&A using MindJourney.**

