# OpenReview forum: "MindJourney: Test-Time Scaling with World Models for Spatial Reasoning"
_NeurIPS.cc/2025/Conference — NeurIPS 2025 poster_

### Official Review · Reviewer_w3Uk · 2025-06-22

**Clarity:** 4
**Significance:** 2
**Originality:** 2
**Rating:** 4
**Confidence:** 4

**Summary:**

This paper proposes SpatialNavigator, a test-time framework that enhances vision-language models' (VLMs) spatial reasoning by coupling them with a controllable video diffusion world model. Without requiring fine-tuning, the method enables VLMs to iteratively explore 3D scenes by planning camera trajectories while the world model generates corresponding synthetic views; these imagined observations are then evaluated and cached to help answer spatial reasoning questions. Experiments across four VLMs and two world models demonstrate consistent improvements (average +8.1% accuracy on the SAT benchmark), showing that this plug-and-play approach complements—and sometimes surpasses—existing methods like RL-finetuned VLMs, while revealing limitations in long-horizon scene consistency that suggest directions for future world model development.

**Questions:**

1. **World Model Quality**: What are the FVD/LPIPS scores for generated views at 1/3/5 steps? Include metrics for both indoor and outdoor scenes.

2. **Failure Analysis**: What are the top 3 failure modes of your world model (e.g., object disappearance)? What percentage of errors do they account for?

3. **Compute Efficiency**: What is the end-to-end latency for SAT questions (breakdown: world model calls vs VLM scoring)? How does it scale with beam width B?

4. **Real-World Generalization**: Can you show results on real embodied tasks (ScanNet in Thinking in Space)?

5. What is the human performance gap on SAT after your improvements? What errors remain?

**Ethical Concerns:**

["NO or VERY MINOR ethics concerns only"]

**Final Justification:**

The response sufficiently addressed the concerns I raised in the initial review, particularly the technical novelty and missing baselines. I find the clarifications satisfactory and have updated my score to reflect this.

**Limitations:**

yes

**Quality:**

4

**Strengths And Weaknesses:**

## Strengths

### 1. Technical Soundness
The integration of video diffusion models (for 3D consistency) with VLM-guided search is well-executed. The beam search formulation is reasonable and addresses combinatorial explosion in trajectory space. Pipeline (Fig. 2) and algorithm (Alg. 1) are clearly explained. Key components (world model formulation, action space, VLM scoring) are detailed in separate subsections.

### 2. Practical Impact
Addresses a critical limitation of VLMs (static 2D reasoning) for embodied reasoning. The plug-and-play nature makes it accessible for real-world deployment.

### 3. Rigorous Evaluation
Extensive experiments on the SAT benchmark (synthetic + real images) with multiple VLMs (GPT-4o, o1, etc.) and world models (SWM, SVC), demonstrating consistent gains (+8.1% avg.). Ablation studies validate design choices.

---

## Weaknesses

### 1. Clarify Technical Novelty
The proposed use of a video world model for test-time scaling in spatial reasoning appears incremental, as conditioning video diffusion on camera poses builds directly on prior works (e.g., CamCtrl, Stable-Virtual-Camera). The core novelty seems to lie in the *interaction* between the VLM and the world model rather than the individual components. The authors should explicitly highlight and differentiate their key technical contributions to avoid misperception of triviality.

### 2. Address World Model Limitations
- **Long-horizon consistency**: Breakdowns in multi-step reasoning (e.g., Fig. 4a) suggest the approach is constrained by current video diffusion models’ limitations. A deeper analysis of failure modes would strengthen the discussion.
- **Quantitative evaluation**: Metrics like FVD or LPIPS for view synthesis quality are absent. Including such analyses would provide a more rigorous assessment of the world model’s capabilities.

### 3. Analyze Computational Trade-offs
The test-time search requires multiple world model calls, but the computational costs (latency, throughput, or energy usage) are not discussed. A brief analysis of scalability—especially for real-world applications—would help readers understand practical limitations.

### 4. Expand Evaluation Scope
The focus on the synthetic SAT benchmark limits the generality of claims. Evaluating on additional spatial reasoning tasks—such as real-world 3D scene understanding (e.g., [1]) or multi-view consistency (e.g., [2])—would better demonstrate the method’s robustness and broader applicability.

### 5. Include Missing Baselines
Comparisons with alternative approaches are needed to contextualize the benefits of test-time scaling:
- **3D-aware VLMs**: How does the method compare to architectures explicitly designed for spatial reasoning?
- **End-to-end fine-tuning**: Does test-time adaptation outperform direct model adjustments?


[1] Yang, J., Yang, S., Gupta, A.W., Han, R., Fei-Fei, L. and Xie, S., 2025. Thinking in space: How multimodal large language models see, remember, and recall spaces. In Proceedings of the Computer Vision and Pattern Recognition Conference (pp. 10632-10643).

[2] Yeh, C.H., Wang, C., Tong, S., Cheng, T.Y., Wang, R., Chu, T., Zhai, Y., Chen, Y., Gao, S. and Ma, Y., 2025. Seeing from another perspective: Evaluating multi-view understanding in mllms. arXiv preprint arXiv:2504.15280.

---

> ### Author Rebuttal · Authors · 2025-07-31
>
> Dear Reviewer,
>
> We thank you for the detailed comments and questions. We are glad to hear you find our method "well-executed and reasonable" and our evaluation "rigorous". We address each concern below:
>
> ### Technical Novelty
> We would like to clarify that our contribution does not lie in proposing a new video diffusion model, but rather in the test-time collaboration protocol that integrates a frozen VLM with a pose‑conditioned world model to enhance 3D spatial reasoning:
>
> **VLM‑guided imagination.** SpatialNavigator, to our knowledge, is the first framework in which a VLM actively controls a world model at inference time to bolster its own reasoning.
>
> **Beam‑search coupling.** The VLM/world-model loop is cast as beam search, balancing diversity and evidence scoring.
>
> We will revise the introduction sections to highlight that our novelty lies in this inference-time coordination mechanism, which builds on, but is distinct from prior works using camera-conditioned video models.
>
> ### World Model Limitations
> We have already included additional analysis of the quality of our trained world model and SVC, and also their failure modes in the supplementary material (Sec. B).
> For the former, using the out-of-domain simulator AI2Thor, we evaluated the world model's fidelity under a sequence of atomic actions defined in our work. We report the PSNR, LPIPS, and SSIM in Tab. 1 of the supplemental. The SVC and SWM models obtain comparable SSIM scores but the latter outperforms the former by larger margins of ~2 and 0.18 for PSNR and SSIM. We refer all reviewers to Sec. B.1.1 for more information.
> To further analyze the reliance of these world models, we evaluate the performance when extended to multiple steps(1 to 3) in the following table.
> ||PSNR$\uparrow$|SSIM$\uparrow$|LPIPS$\downarrow$|
> |-|-|-|-|
> |SVC-step1|67.38|0.997|0.39|
> |SVC-step2|63.98|0.994|0.54|
> |SVC-step3|62.80|0.993|0.57|
> |SWM-step1|67.35|0.998|0.26|
> |SWM-step2|66.71|0.997|0.30|
> |SWM-step3|65.98|0.997|0.35|
>
> Generally, we observe consistent trends across the SVC and SWM models, where their effectiveness gradually decrease as the number of steps increases. This is expected since the uncertainty of predicting the exact visual changes is compounded over steps.
>
> Despite the current limitations of learned world models, SpatialNavigator consistently boosts VLM reasoning performance. Ongoing advances—illustrated by lighter, higher-fidelity backbones such as Wan2.1 and the controllable ReCamMaster—provide a clear trajectory for further progress. As these models mature, we expect them to unlock even greater downstream performance with SpatialNavigator.
>
> ### Expand Evaluation Scope
> With regards to expanding the evaluation scope to real 3D scenes, we have also demonstrated the effectiveness of our method on SAT-Real in Tab. 1 of the main paper, where it achieves performance gains consistent with prior observations. As additional evidence of the generalizability of our approach, we also provide evaluation results on MindCube, which tasks VLMs to perform multiview spatial reasoning in real 3D scenes. We only evaluated with SVC, as it supports video generation conditioned on multiview images, unlike SWM which currently lacks this capability.
> ||Acc.|
> |-|-|
> |gpt4o|44.0|
> |+ SN (SVC)|51.3|
> |gpt4.1|43.3|
> |+ SN (SVC)|47.3|
>
> Unlike static and single-view image tasks, the multiview setting in MindCube requires to understand 3D spatial layouts and cross-view consistency. The results further show the potential of our method in real-world and multiview settings. We observe that SpatialNavigator is able to improve accuracy by approximately 4 to 7% across different VLMs. Thus, SpatialNavigator enables VLMs to generalize to multiview settings by integrating visual cues from multiple imagined perspectives. This capability highlights its potential in embodied AI and robotic tasks.
> Finally, the results suggest that SpatialNavigator is model-agnostic and can further improve with advances in better world models as well as VLMs. We will include these results in the next version of the paper to make our evaluations more comprehensive.
>
>
> ### Computational Efficiency
> We appreciate the reviewer’s suggestion and agree that analyzing computational trade-offs is important for practical deployment. To address this, we provide a structured analysis of the current computational efficiency of SpatialNavigator.
>
> Also, we would like to clarify that SpatialNavigator is the first work on test-time scaling framework that uses VLM to control a video diffusion model. Thus, we mainly focused on optimizing the effectiveness instead of efficiency, we acknowledge that efficiency optimization is a promising direction for future work. To address this, we outline several acceleration techniques that make further speed-ups highly plausible.
>
> #### Efficiency Analysis
> We profiled computational efficiency on a single H100-80G. We evaluated using gpt4o+SN(SWM), with beam size 2 and inference step 50 on SAT-Real. The memory consumption is~40G across the experiments. As SVC cam only generates high resolution video and therefore not efficient(around x3 world model latency), we mainly use SWM in this section.
> |Model|Step|SWM time(s)|vlm_search time(s)|vlm_qa time(s)|total time(s)|acc%|
> |-|-|-|-|-|-|-|
> |gpt4o|0|-|-|1.57|1.57|60.0|
> |gpt4o|1|14.52|4.70|1.69|20.91|66.0|
> |gpt4o|2|39.53|8.40|1.85|49.78|70.7|
> |gpt4o|3|52.78|13.53|3.35|69.66|68.7|
>
> In our main experiments, we focused on optimizing the effectiveness so the default setting is not the most cost-effective setting. To explore more cost-optimized settings, we further investigate reducing the number of inference step in the table below:
>
> |Iters|Step|SWM time(s)|vlm_search time(s)|vlm_qa time(s)|total time(s)|acc%|
> |-|-|-|-|-|-|-|
> |10|1|4.66|4.66|2.56|11.88|66.7|
> |10|2|7.62|8.28|2.72|18.62|66.7|
> |10|3|10.69|11.25|5.79|27.73|67.3|
> |20|1|7.22|4.44|2.48|14.14|66.0|
> |**20**|**2**|11.76|9.14|2.50|**23.40**|**68.0**|
> |20|3|23.51|10.30|4.29|38.10|67.3|
>
> The results demonstrate that our model can maintain a >95% of the accuracy gain at ≤33% of the latency, with a latency of about 23s. We tested o1 on the SAT-Real with latency in average 19.6s, so SpatialNavigator in cost-optimized setting is in the same latency band as widely used test-time-scaling pipelines.
>
> We further added the ablation study on beam size in the table below:
> |Model|Step|Beam|SWM time(s)|vlm_search time(s)|vlm_qa time(s)|total time(s)|acc%|
> |-|-|-|-|-|-|-|-|
> |gpt4o|2|1|23.24|7.51|1.60|32.35|67.3|
> |gpt4o|2|2|39.53|8.40|1.85|49.78|70.7|
> |gpt4o|3|1|32.16|14.43|3.58|50.17|68.0|
> |gpt4o|3|2|52.78|13.53|3.35|69.66|68.7|
>
> #### Scalability & Acceleration Pathways
>
> **Backbone Model**
> Recent released Wan2.1-1.3B provides videos with quality higher than our backbone CogVideoX-5B in ≈8GB VRAM. Wan2.1 achieved ≈1.7×lower (8.7s vs.14.5s) latency and ≈½ the memory footprint relative our current backbone tested under identical frame counts and resolution,
>
> **Step Distillation**
> Latent Consistency Models produce high-fidelity outputs in 2–4 inference steps, yielding 10–20× speed-ups over 50-step schedulers. Video-specific extensions such as Distribution-Matching Distillation report similar benefits on text-to-video transformers.
>
> **Low-Bit Quantization**
> Post-training methods like SVDQuant quantize both weights and activations to 4-bit while keeping FID within ±2 points of full precision. Empirical results report ≈3× speedups and up to 3.6× memory reduction on DiTs, enabling real-time sampling on a 16GB laptop-class GPU.
>
> All three techniques are open-source and multually compatible, which indicates that improving computational cost is both practical and well-supported by the community.
>
>
> ### Missing Baselines
> For additional baselines, we report results for (i)a recent 3D-aware VLM VLM3R [1] and (ii)end-to-end fine-tuning on SAT-Real/SAT-Synth.
>
> **3D-aware VLMs**
> VLM3R was evaluated under the same zero-shot protocol used for our method (SpatialNavigator, SN).
> |Accuracy|SAT-Real|SAT-Synth.|
> |-|-|-|
> |gpt4o|0.603|0.61|
> |+ SN (SWM)|0.68|0.708|
> |VLM3R|0.527|0.512|
>
> Despite being explicitly designed for 3D spatial reasoning, VLM3R lags behind both the GPT-4o baseline and our test-time scaling scheme, underscoring the effectiveness of coupling a strong VLM with a controllable world model at inference time.
>
> **End-to-End Fine-Tuning**
> Fine-tuning requires task-specific training data, whereas our approach is strictly zero-shot, which makes the comparison unfair. Nevertheless, for context we refer the finetuning baselines from the SAT paper with our numbers:
> |Acc.|SAT-Real|Improvements|
> |-|-|-|
> |LLaVA-1.5-13B|41.6|-|
> |+ Finetuned|54.9|+13.3|
> |LLaVA-Video-7B|53.5|-|
> |+ Finetuned|63.4|+9.9|
> |gpt4o|60.6|-|
> |+ SN(SVC)|69.3|+8.7|
>
> The absolute gains delivered by test-time scaling(+8.7%) are on par with supervised fine-tuning(+9~13%), while preserving the zero-shot assumption and avoiding additional training costs.
> [1] Z, Fan., et al. VLM-3R: Vision-Language Models Augmented with Instruction-Aligned 3D Reconstruction. arXiv preprint, 2025.
>
> ### Other Questions
>
> **World Model quality in outdoor scenes**
> Assessing world-model fidelity outdoors would require using a dedicated outdoor simulator to render GT image sequences for atomic actions, which require engineering efforts to set up. As the outdoor environments are inherently unstructured (e.g., vegetation, variable lighting), generalizing to outdoor scene will be comparatively challenging. Although SpatialNavigator also boost performance on SAT-Real with outdoor scenes, this study targets more on structured indoor domain.
>
> **Top 3 Failure Mode**
> We manually inspected part of the rollouts and the top 3 failure modes are B.1.1 Inaccurate Forward Movement, B.1.4 Visual Artifacts, and B.1.5 Scene Misinterpretation mentioned in the supplimentary material.
>
> **Human Performance Gap**
> Humans score 92.8% on SAT-Synth.—≈15% above our best model—signaling significant headroom for future work.

---

> > ### Comment · Reviewer_w3Uk · 2025-08-06
> >
> > The response sufficiently addressed the concerns I raised in the initial review, particularly the **technical novelty** and **missing baselines**. I find the clarifications satisfactory and have updated my score to reflect this.

---

### Official Review · Reviewer_Pkry · 2025-06-30

**Clarity:** 3
**Significance:** 2
**Originality:** 3
**Rating:** 5
**Confidence:** 4

**Summary:**

This paper proposes a novel spatially-aware beam search algorithm that allows to improve the performance of existing VLMs at test-time by coupling a VLM with a posed-controlled diffusion models that acts as a world model.

**Questions:**

1. Table 1 (SAT-Real).
    * Why are GPT-4.1 + SN(SVC) and InternVL + SN(SVC) missing?


2. World-model quality.
    * Can you quantify how the DM quality correlates with downstream accuracy? (An experiment where you degrade SWM (e.g. an early checkpoint) or compare longer roll-outs would help.)
    * Could you provide a simple heuristic or automatic self-check so practitioners know when their world-model is “good enough” for the presented methodology?


3. Runtime.
    * Please report the average wall-clock time per VQA  and compare it to the vanilla VLM baseline.
    * A plot of accuracy vs. inference-time budget would let readers judge whether the reported  gain justifies the extra compute.

4. Hyper-parameter sensitivity.
    * How robust is the algorithm  to the thresholds  ?
    * What happens if you restrict the action space to {move-forward} only but allow deeper search, or conversely allow all three actions but only one step?

**Ethical Concerns:**

["NO or VERY MINOR ethics concerns only"]

**Final Justification:**

I think the response addresses my concerns. I recommend to incorporate all new experiments to the paper. I feel this is an interesting direction and I feel positive about acceptance of this paper.

**Limitations:**

yes

**Quality:**

3

**Strengths And Weaknesses:**

Strengths
* The idea of “adding” a visual/spatial dimension to beam search is thought-provoking. One can imagine extending the same principle to other search algorithms.
* Experiments are fairly comprehensive: the authors evaluate on both proprietary  and open-source  VLMs, across synthetic and real SAT benchmarks, and with two distinct world models (their SWM and SVC). I particularly appreciate showing SVC even tho the results on o1 are not as good.
* Reported gains are consistent.


Weaknesses:
* Inference cost. Even vanilla beam search is slow; adding a video-diffusion rollout at every node seems expensive for simple VQA tasks. The paper gives no wall-clock numbers or cost–accuracy plots, so practicality remains unclear.
* Single-view assumption. The pipeline assumes exactly one reference image; the authors list this as a limitation themselves . It is unclear how SBS would handle multiple views or a short video.
* Reliance on world-model fidelity. The approach assumes the pose-conditioned diffusion model can render a faithful scene. In practice, a single image underspecifies geometry, so hallucinations could hurt more than help. The authors also acknowledge this,.
* Scalability knobs left unexplored. Apart from the brief ablation, the paper mostly fixes one search configuration for all experiments . It would be useful to see accuracy vs. (a) action-set size and (b) beam depth to understand the compute/accuracy trade-off. For instance, what’s the performance if the action space is reduced to only one action vs two, and the depth of the tree is increased?


Overall, the idea is interesting and the empirical gains seem solid, but the paper needs more discussion (and data) on computational cost and on when the world-model may fail.

---

> ### Author Rebuttal · Authors · 2025-07-31
>
> Dear Reviewer Pkry,
>
> We appreciate your detailed comments and insightful suggestions. We are glad to hear you find our idea "thought-provoking" and our experiments "comprehensive". Here are the responses to your questions.
>
> ### Computational Efficiency
> We appreciate the reviewer’s suggestion and agree that analyzing computational trade-offs is important for practical deployment. To address this, we provide a structured analysis of the current computational efficiency of SpatialNavigator.
>
> Also, we would like to clarify that SpatialNavigator is the first work on test-time scaling framework that uses VLM to control a video diffusion model. Thus, we mainly focused on optimizing the effectiveness instead of efficiency, we acknowledge that efficiency optimization is a promising direction for future work. To address this, we outline several acceleration techniques that make further speed-ups highly plausible.
>
> #### Efficiency Analysis
> We profiled computational efficiency on a single H100-80G. We evaluated using gpt4o+SN(SWM), with beam size 2 and inference step 50 on SAT-Real. The memory consumption is~40G across the experiments. As SVC cam only generates high resolution video and therefore not efficient(around x3 world model latency), we mainly use SWM in this section.
> |Model|Step|SWM time(s)|vlm_search time(s)|vlm_qa time(s)|total time(s)|acc%|
> |-|-|-|-|-|-|-|
> |gpt4o|0|-|-|1.57|1.57|60.0|
> |gpt4o|1|14.52|4.70|1.69|20.91|66.0|
> |gpt4o|2|39.53|8.40|1.85|49.78|70.7|
> |gpt4o|3|52.78|13.53|3.35|69.66|68.7|
>
> In our main experiments, we focused on optimizing the effectiveness so the default setting is not the most cost-effective setting. To explore more cost-optimized settings, we further investigate reducing the number of inference step in the table below:
>
> |Iters|Step|SWM time(s)|vlm_search time(s)|vlm_qa time(s)|total time(s)|acc%|
> |-|-|-|-|-|-|-|
> |10|1|4.66|4.66|2.56|11.88|66.7|
> |10|2|7.62|8.28|2.72|18.62|66.7|
> |10|3|10.69|11.25|5.79|27.73|67.3|
> |20|1|7.22|4.44|2.48|14.14|66.0|
> |**20**|**2**|11.76|9.14|2.50|**23.40**|**68.0**|
> |20|3|23.51|10.30|4.29|38.10|67.3|
>
> The results demonstrate that our model can maintain a >95% of the accuracy gain at ≤33% of the latency, with a latency of about 23s. We tested o1 on the SAT-Real with latency in average 19.6s, so SpatialNavigator in cost-optimized setting is in the same latency band as widely used test-time-scaling pipelines.
>
> We further added the ablation study on beam size in the table below:
> |Model|Step|Beam|SWM time(s)|vlm_search time(s)|vlm_qa time(s)|total time(s)|acc%|
> |-|-|-|-|-|-|-|-|
> |gpt4o|2|1|23.24|7.51|1.60|32.35|67.3|
> |gpt4o|2|2|39.53|8.40|1.85|49.78|70.7|
> |gpt4o|3|1|32.16|14.43|3.58|50.17|68.0|
> |gpt4o|3|2|52.78|13.53|3.35|69.66|68.7|
>
> #### Scalability & Acceleration Pathways
>
> **Backbone Model**
> Recent released Wan2.1-1.3B provides videos with quality higher than our backbone CogVideoX-5B in ≈8GB VRAM. Wan2.1 achieved ≈1.7×lower (8.7s vs.14.5s) latency and ≈½ the memory footprint relative our current backbone tested under identical frame counts and resolution.
>
> **Step Distillation**
> Latent Consistency Models produce high-fidelity outputs in 2–4 inference steps, yielding 10–20× speed-ups over 50-step schedulers. Video-specific extensions such as Distribution-Matching Distillation report similar benefits on text-to-video transformers.
>
> **Low-Bit Quantization**
> Post-training methods like SVDQuant quantize both weights and activations to 4-bit while keeping FID within ±2 points of full precision. Empirical results report ≈3× speedups and up to 3.6× memory reduction on DiTs, enabling real-time sampling on a 16GB laptop-class GPU.
>
> All three techniques are open-source and multually compatible, which indicates that improving computational cost is both practical and well-supported by the community.
>
> ### Multiview Evaluation
> As additional evidence of the generalizability of our approach, we provide evaluation results on the MindCube benchmark, which evaluates VLMs' capability to perform multiview spatial reasoning in real 3D scenes. We used the Stable Virtual Camera (SVC) backbone because SVC can synthesize videos conditioned on multiple input views, whereas our current SWM lacks multiview generation support. MindCube supplies an initial frame for every question, enabling us to plug our SpatialNavigator into the multiview setting with only minor modifications.
>
> ||Acc.|
> |-|-|
> |gpt4o|44.0|
> |+ SN (SVC)|51.3|
> |gpt4.1|43.3|
> |+ SN (SVC)|47.3|
>
> Unlike static and single-view image datasets, the multiview setting in MindCube requires the ability to understand 3D spatial layouts and cross-view consistency.
> The results further demonstrate the potential of our method in real-world and multiview settings. We observe that SpatialNavigator is able to improve average accuracy by approximately 4 to 7% across different VLMs. Thus, SpatialNavigator enables VLMs to generalize to multiview settings by integrating visual cues from multiple imagined perspectives. This capability highlights its potential in embodied AI and robotic tasks. Last but not least, the results suggest that SpatialNavigator is model-agnostic and can further improve with advances in more capable world models as well as VLMs. We will include these results in the next version of the paper to make our evaluations more comprehensive.
>
>
> ### World Model Limitations
> We included additional analysis of the quality of SWM and SVC, and also their failure modes in the supplementary material, which helps the community to quantify the world model quality in our application and understand the weakness of current world models for further improvement (Sec B).
>
> For the former, using the out-of-domain simulator AI2Thor, we evaluated the world model's fidelity under a sequence of atomic actions (turn-left/right, move-forward) defined in our work. We report the PSNR, LPIPS, and SSIM in Table 1 of the supplemental. The SVC and SWM models obtain comparable SSIM scores but the latter outperforms the former by larger margins of ~2 and 0.18 for the PSNR and SSIM scores. We refer all reviewers to Section B.1.1 of our supplemental for more information.
>
> #### Quanitifying World Model Quality vs. Downstream Performance
>
> To gauge how visual quality translates into reasoning, we ran SpatialNavigator with the original SWM and two deliberately degraded checkpoints.
> |  | PSNR $\uparrow$| SSIM $\uparrow$| LPIPS(1e-4) $\downarrow$ | SAT-Real | SAT-Synthesized |
> | -------- | -------- |  -------- | -------- |  -------- | -------- |
> | SWM-original | 66.59 $\pm$ 0.21| 0.997 $\pm$ 0.001| 0.31 $\pm$ 0.02  | 68.0 | 70.8 |
> | SWM-checkpoint1 | 65.06 $\pm$ 0.28| 0.996 $\pm$ 0.001| 0.44 $\pm$ 0.02  | 63.3 | 66.0 |
> | SWM-checkpoint2 | 64.03 $\pm$ 0.28| 0.995 $\pm$ 0.001| 0.50 $\pm$ 0.03  | 61.3 | 63.0 |
>
> As PSNR/SSIM drop and LPIPS rises, SAT accuracy falls proportionally, showing that the three metrics, especially PSNR and LPIPS provide a practical heuristic for evaluating world models before downstream deployment.
>
> ### More Ablation Studies and Experiments
>
>
> #### Experiments
> | Config        | Avg  | EgoM  | ObjM | EgoAct | GoalAim | Pers |
> | ------------- | ---- | ----- | ---- | ------ | ------- | ---- |
> | GPT-4.1+SVC   | 77.3 | 100.0 | 82.6 | 86.5   | 79.4    | 45.4 |
> | Internvl3+SVC | 66.7 | 69.6  | 60.9 | 78.4   | 79.4    | 42.4|
>
> We added the experiments on gpt-4.1 + SVC and InterVL3 + SVC on the SAT-Real as suggested by the reviewer
> #### Action Space Ablations
> To disentangle the role of translation versus rotation in viewpoint acquisition, we evaluated two reduced-action variants of our policy: (i) No-Move—all move-forward commands were removed; (ii) No-Turn—all turn-left / turn-right commands were removed.
> | Config | Acc. |
> | -------- | -------- |
> | GPT-4o      |70.8|
> | GPT-4o+forward_only | 64.6 |
> | GPT-4o+Turn_only | 69.3 |
>
> According to the ablation results, the performance of turn_only is close to the complete action set. Rotations are potentially more important than forward for the tasks in SAT.
>
> #### Threshold Ablations
> Section 4.3 varies the pruning threshold τ controlling which roll-outs are retained. As illustrated in Fig. 4, very low thresholds (τ ≤ 5) admit numerous low-fidelity frames; accumulated error then grows with trajectory length. Performance stabilises once τ ≥ 7—the curves for τ = 7 and τ = 9 are similar.

---

> > ### Comment · Reviewer_Pkry · 2025-08-05
> >
> > I thank the authors for their comprehensive response. I think the response addresses my concerns. I feel this is an interesting direction and I feel positive about acceptance of this paper.

---

### Official Review · Reviewer_27Ro · 2025-07-03

**Clarity:** 2
**Significance:** 2
**Originality:** 3
**Rating:** 2
**Confidence:** 5

**Summary:**

Summary
This paper introduces SpatialNavigator, a test-time scaling framework designed to enhance the spatial reasoning capabilities of vision-language models (VLMs) by pairing them with action-driven, pose-conditioned video world models. The pipeline operates by letting the VLM iteratively plan exploratory camera trajectories, which the world model renders into imagined observations. These are then used by the VLM to answer spatial-reasoning queries, leveraging multi-view synthetic evidence at inference time without any additional training.

**Questions:**

See weakness

**Ethical Concerns:**

["NO or VERY MINOR ethics concerns only"]

**Final Justification:**

After reading the rebuttal and reviewing the provided case studies and videos, my main takeaway is that the inclusion of the world model brings negligible benefit to spatial reasoning, and in some cases, may even be unnecessary or redundant.

First, **zero-shot generalization in video generation is inherently very limited**. When the viewpoint shift involves a significant rotation or movement, the model must hallucinate content that was not clearly visible in the first frame. This makes the generated frames unreliable, especially for reasoning tasks that demand accurate spatial understanding. **This limitation is clearly visible in the authors’ videos and qualitative results** — the world model is only capable of generating scenes under minor rotations or small translations, which aligns with current known limitations of such models.

Second, **the generation process takes several seconds**, which raises serious concerns about practicality and scalability. If the spatial reasoning gain is has limited application and unstable, it is unclear what value the additional computational burden brings to the overall pipeline.

In summary, the proposed use of the world model seems to add substantial complexity and latency while providing minimal actual benefit to spatial reasoning or downstream performance.

**Limitations:**

Yes

**Quality:**

2

**Strengths And Weaknesses:**

Strengths And Weaknesses

Strengths:
1. The paper identifies a core challenge in spatial reasoning for VLMs, emphasizing the inability of current models to mentally simulate 3D scene dynamics from 2D inputs.

2. The strategy of coupling an off-the-shelf VLM with a video world model is systematic and well grounded.

3. This paper is well-written and well-organized.

Weaknesses:
1. The performance of the entire system appears to heavily depend on the quality of the world model.

2. The test-time computation required for multiple simulated rollouts, per-step VLM scoring, and evidence buffer construction could be substantial, especially for deeper search settings and larger action spaces

3. Although the method is compared to some RL-based TTS VLMs, more explicit analysis or comparison to recent test-time ensembling, chain-of-thought reranking, verifier-based TTS, or model-based counterfactual data augmentation could provide a clearer picture of strengths relative to the broader TTS design space.

---

> ### Author Rebuttal · Authors · 2025-07-31
>
> Dear Reviewer,
>
> We thank the reviewer for the detailed reviews and insightful suggestions. We are glad to hear that you find our method "systematic and well-grounded" and our paper well motivated and well-written. Here are the responses to your questions.
>
> ### World Model Limitations
> We acknowledge that the performance of our method depends on the fidelity of the world models. Therefore, we provide metrics and visualizations to analyze this dependency both quantitatively and qualitatively. We included additional analysis of the quality of SWM and SVC, and also their failure modes in the supplementary material, which helps the community to quantify the world model quality in our application and understand the weakness of current world models for further improvement (Sec B).
>
> For the former, using the out-of-domain simulator AI2Thor, we evaluated the world model's fidelity under a sequence of atomic actions (turn-left/right, move-forward) defined in our work. We report the PSNR, LPIPS, and SSIM in Table 1 of the supplemental. The SVC and SWM models obtain comparable SSIM scores but the latter outperforms the former by larger margins of ~2 and 0.18 for the PSNR and SSIM scores. We refer all reviewers to Section B.1.1 of our supplemental for more information.
>
> #### Quanitifying World Model Quality vs. Downstream Performance
>
> To gauge how visual quality translates into reasoning, we ran SpatialNavigator with the original SWM and two deliberately degraded checkpoints.
> |  | PSNR $\uparrow$| SSIM $\uparrow$| LPIPS(1e-4) $\downarrow$ | SAT-Real | SAT-Synthesized |
> | -------- | -------- |  -------- | -------- |  -------- | -------- |
> | SWM-original | 66.59 $\pm$ 0.21| 0.997 $\pm$ 0.001| 0.31 $\pm$ 0.02  | 68.0 | 70.8 |
> | SWM-checkpoint1 | 65.06 $\pm$ 0.28| 0.996 $\pm$ 0.001| 0.44 $\pm$ 0.02  | 63.3 | 66.0 |
> | SWM-checkpoint2 | 64.03 $\pm$ 0.28| 0.995 $\pm$ 0.001| 0.50 $\pm$ 0.03  | 61.3 | 63.0 |
>
> As PSNR/SSIM drop and LPIPS rises, SAT accuracy falls proportionally, showing that the three metrics, especially PSNR and LPIPS provide a practical heuristic for evaluating world models before downstream deployment.
>
> #### Future Development of World Models
>
> Despite the current limitations of learned world models, SpatialNavigator consistently boosts VLM reasoning performance. Ongoing advances—illustrated by lighter, higher-fidelity backbones such as Wan2.1 and the controllable ReCamMaster—provide a clear trajectory for further progress. As these models improve, we expect them to unlock even greater downstream performance with SpatialNavigator.
>
> ### Computational Efficiency
> We appreciate the reviewer’s suggestion and agree that analyzing computational trade-offs is important for practical deployment. To address this, we provide a structured analysis of the current computational efficiency of SpatialNavigator.
>
> Also, we would like to clarify that SpatialNavigator is the first work on test-time scaling framework that uses VLM to control a video diffusion model. Thus, we mainly focused on optimizing the effectiveness instead of efficiency, we acknowledge that efficiency optimization is a promising direction for future work. To address this, we outline several acceleration techniques that make further speed-ups highly plausible.
>
> #### Efficiency Analysis
> We profiled computational efficiency on a single H100-80G. We evaluated using gpt4o+SN(SWM), with beam size 2 and inference step 50 on SAT-Real. The memory consumption is~40G across the experiments. As SVC cam only generates high resolution video and therefore not efficient(around x3 world model latency), we mainly use SWM in this section.
> |Model|Step|SWM time(s)|vlm_search time(s)|vlm_qa time(s)|total time(s)|acc%|
> |-|-|-|-|-|-|-|
> |gpt4o|0|-|-|1.57|1.57|60.0|
> |gpt4o|1|14.52|4.70|1.69|20.91|66.0|
> |gpt4o|2|39.53|8.40|1.85|49.78|70.7|
> |gpt4o|3|52.78|13.53|3.35|69.66|68.7|
>
> In our main experiments, we focused on optimizing the effectiveness so the default setting is not the most cost-effective setting. To explore more cost-optimized settings, we further investigate reducing the number of inference step in the table below:
>
> |Iters|Step|SWM time(s)|vlm_search time(s)|vlm_qa time(s)|total time(s)|acc%|
> |-|-|-|-|-|-|-|
> |10|1|4.66|4.66|2.56|11.88|66.7|
> |10|2|7.62|8.28|2.72|18.62|66.7|
> |10|3|10.69|11.25|5.79|27.73|67.3|
> |20|1|7.22|4.44|2.48|14.14|66.0|
> |**20**|**2**|11.76|9.14|2.50|**23.40**|**68.0**|
> |20|3|23.51|10.30|4.29|38.10|67.3|
>
> The results demonstrate that our model can maintain a >95% of the accuracy gain at ≤33% of the latency, with a latency of about 23s. We tested o1 on the SAT-Real with latency in average 19.6s, so SpatialNavigator in cost-optimized setting is in the same latency band as widely used test-time-scaling pipelines.
>
> We further added the ablation study on beam size in the table below:
> |Model|Step|Beam|SWM time(s)|vlm_search time(s)|vlm_qa time(s)|total time(s)|acc%|
> |-|-|-|-|-|-|-|-|
> |gpt4o|2|1|23.24|7.51|1.60|32.35|67.3|
> |gpt4o|2|2|39.53|8.40|1.85|49.78|70.7|
> |gpt4o|3|1|32.16|14.43|3.58|50.17|68.0|
> |gpt4o|3|2|52.78|13.53|3.35|69.66|68.7|
>
> #### Scalability & Acceleration Pathways
>
> **Backbone Model**
> Recent released Wan2.1-1.3B provides videos with quality higher than our backbone CogVideoX-5B in ≈8GB VRAM. Wan2.1 achieved ≈1.7×lower (8.7s vs.14.5s) latency and ≈½ the memory footprint relative our current backbone tested under identical frame counts and resolution.
>
> **Step Distillation**
> Latent Consistency Models produce high-fidelity outputs in 2–4 inference steps, yielding 10–20× speed-ups over 50-step schedulers. Video-specific extensions such as Distribution-Matching Distillation report similar benefits on text-to-video transformers.
>
> **Low-Bit Quantization**
> Post-training methods like SVDQuant quantize both weights and activations to 4-bit while keeping FID within ±2 points of full precision. Empirical results report ≈3× speedups and up to 3.6× memory reduction on DiTs, enabling real-time sampling on a 16GB laptop-class GPU.
>
> All three techniques are open-source and multually compatible, which indicates that improving computational cost is both practical and well-supported by the community.
>
>
> ### Test-Time Scaling Baselines
>
> As suggested by the reviewer, we implemented two classic test-time scaling baselines for further comparisons.
> 1) Self-Consistency: The model is queried n times to give answers and rationales; the majority answer is returned.
> 2) Re-Ranking: The model generates n rationales and then scores them itself; the top-scoring answer is chosen.
>
> | Config | SAT_test | SAT_val |
> | -- | --- |  -- |
> | gpt-4o | 60.3|61.0|
> | + SN (SWM) |68.0|70.8|
> | + SN (SVC) |69.3|72.3|
> | + Self-Consistency 10 samples | 61.3|62.2 |
> | + Self-Consistency 30 samples | 63.3|61.4 |
> | + Re-Ranking 10 samples | 62.7| 63.6|
> | + Re-Ranking 30 samples | 59.3| 63.2|
>
> We evaluated the two methods using gpt-4o as the base model and experimented with different n values, 10 and 30.
>
> According the tables, although the classic test-time scaling methods generally improve from the baselines, they are not as effective as SpatialNavigators for the spatial reasoning tasks.

---

> > ### Comment · Reviewer_27Ro · 2025-08-07
> >
> > Thank you for the clarifications and additional results.
> >
> > 1. I remain skeptical of the statement "The resulting mix couples Habitat’s geometric fidelity with the visual diversity of real indoor and outdoor scenes, allowing the world model to generalize beyond its synthetic training domain." Based on my experience, it is very difficult to train a reliable and generalizable world model with only 50K HM3D samples and 20K real videos. The evaluation is only conducted on the Spatial Aptitude Training (SAT) benchmark, a relatively uncommon simulation-based dataset, which I do not find sufficient to support the idea.  Why didn’t you consider alternative standard 3D spatial QA datasets？
> >
> > 2. The qualitative results shown in the video mostly depict very small movements and do not convincingly demonstrate the model’s capabilities. **Also, I feel that this dataset is quite unrealistic**. The observed improvement appears more like a zoom-in effect , rather than a meaningful enhancement in spatial reasoning. Given the substantial cost of training and using such a world model, I am not yet convinced that this is a promising method.

---

> ### Author Response · Authors · 2025-08-07
>
> Thank you for the thoughtful feedback. However, several premises in your comments are inaccurate. We address each point below and provide concrete evidence that directly rebuts the misconceptions.
>
> ### Training Data Size
> > It is very difficult to train a reliable and generalizable world model with only 50K HM3D samples and 20K real videos.
>
> This claim is not consistent with recent controllable video generation model literature.
>
> **Comparison to recent works.** Recent studies show our data regime is well within the range required for high-quality models. Stable Virtual Camera (SVC)[1] is trained on ≈47 K videos, 20 K of which (DL3DV-10K + RealEstate10K) overlap with our corpus. ReCamMaster[2], another camera-controlled video diffusion model, uses 136K videos. Our 70K-video training set therefore lies between these well-established scales.
>
> **Task difficulty.** Unlike ReCamMaster, which must model temporal dynamics, and SVC, which needs to simulate any defined camera trajectories, our Search World Model (SWM) produces **next-view predictions in a static scene** for a **restricted action space**, {move-forward, turn-left, turn-right}, specialized for SpatialNavigator. This narrower scope further lowers the data requirement and training difficulty.
>
> **Strong Backbone Model.** As mentioned at Sec. 3.4, SWM is fine-tuned based on CogVideoX-5B, a powerful video diffusion model. Fine-tuning from a stong backbone model also lowers the data requirement and training difficulty.
>
> Thus, it is feasible to obtain a world model that is reliable enough for our applications with our dataset.
>
> ### Additional Benchmarks
> > Why didn’t you consider alternative standard 3D spatial QA datasets？
>
> We evaluated our method on a subset of a new real-world 3D spatial reasoning benchmark MindCube[3]. The results demonstrate that our method generalizes beyond SAT. (Which also addresses the concerns of reviewer Pkry and w3Uk)
> ||Accuracy|
> |-|-|
> |gpt4o|44.0|
> |+ SN|51.3|
> |gpt4.1|43.3|
> |+ SN|47.3|
>
> ### Realism of the Dataset
> > Also, I feel that this dataset is quite unrealistic.
>
> **SAT-Real.** We have demonstrated the effectiveness of our method on the **real-world** split SAT-Real in Table 1.
>
> **MindCube.** We also demonstrated that our method is effective on a new **real-world** 3D spatial reasoning benchmark MindCube.
>
> ### “Zoom-in” vs. True Spatial Reasoning
> > The improvement appears more like a zoom-in effect, rather than a meaningful enhancement in spatial reasoning.
>
> Our ablation studies decisively refute this explanation.
>
> We evaluated our method on two reduced action space: i) Turn-Only: move-forward is removed from the action space; ii) Forward-Only: turn-left/right are removed. The "Zoom-in Effect" is similar to the Forward-Only setting.
>
> ||SAT-Real|SAT-Synthesized|
> |-|-|-|
> |GPT-4o+full_action_space|70.8|71.0|
> |GPT-4o+forward_only|64.6|63.8|
> |GPT-4o+turn_only|69.3|67.6|
>
> Forward-only—closest to a mere “zoom”—is 6–7% worse than full SpatialNavigator. It demonstrate that **the combination of translation + rotation is important for optimal spatial reasoning**. Therefore, the improvement is not caused by the "zoom-in effect" the reviewer mentioned.
>
> ### Computational Cost
> > Given the substantial cost of training and using a world model, I am not yet convinced that this is a promising method.
>
> **Training Cost.** As we did not train any VLMs, the training cost of SWM is comparable to training VLMs in other related works. Moreover, pre-trained controllable models such as SVC can be used zero-shot, reducing training cost to zero for adopters.
>
> **Inference Cost.** Other than the detailed cost analysis we provided before, we would like to highlight that the **recently released controllable world model, Genie 3 [4], reports real-time rollout**, which illustrates a promising future of our method.
>
> ### Novelty Clarification
> We understand the general concerns of the reviewer on the quality of world models. Although current SWM and SVC are already good enough to provide substantial performance gain, there is indeed room for further improvement, as we also illustrated in the failure modes (Sec. B)
>
> However, our main contribution is **not** improving the controllable world model, but rather in the test-time collaboration protocol that integrates a VLM with a pose‑conditioned world model to enhance 3D spatial reasoning.
>
> Therefore, our method is **orthogonal** to advances in controllable video diffusion, and the new advances like ReCamMaster[2] and Genie 3[4] can **further elevate SpatialNavigator**.
>
> [1] Stable Virtual Camera: Generative View Synthesis with Diffusion Models. https://stable-virtual-camera.github.io/
>
> [2] ReCamMaster: Camera-Controlled Generative Rendering from A Single Video. https://jianhongbai.github.io/ReCamMaster/
>
> [3] MindCube: Spatial Mental Modeling from Limited Views. https://mind-cube.github.io/
>
> [4] Genie 3: A new frontier for world models. https://deepmind.google/discover/blog/genie-3-a-new-frontier-for-world-models/

---

> > ### Comment · Reviewer_27Ro · 2025-08-08
> >
> > Thanks for reply.
> >
> > I do not think the **Comparison to recent works**  is appropriate. These methods are fundamentally designed for camera control-based video generation, **aiming to improve the quality or controllability of generated videos**. In contrast, your work is focused on using video generation as a tool to support downstream spatial reasoning. This is a different problem setting.
> >
> > As for adapting existing video generation models (e.g., Wan2.1-1.3B) to build stronger world models, I would argue that this line of research is still in its early stages. From what I’ve seen so far,  open-sourced finetuned models do not consistently demonstrate reliable spatial understanding. Therefore, using them for test-time enhancement in downstream reasoning tasks seems premature at this point. In my view, the current paper does not convincingly demonstrate that the generative model improves performance on spatial reasoning due to its generation quality. As such, while the direction is interesting, I believe it is too early to draw strong conclusions about its effectiveness for this class of tasks.
> >
> > For zoom-in effect, I mean the current capabilities of your world model are limited to generating minor viewpoint shifts — small camera rotations and translations. The resulting effect is functionally equivalent to a zoom-in or slight reframing of the input scene. While this may indeed help LLMs like GPT4-o perform better by providing more localized or detailed visual context, I would caution against describing this as 3D spatial reasoning with world model.

---

> > > ### Author Response · Authors · 2025-08-08
> > >
> > > We thank the reviewer for acknowledging that the direction we proposed is interesting. However, several points in the review are based on misconceptions, which we address below.
> > >
> > > ### Empirical Performance Improvement
> > > > The current paper does not convincingly demonstrate that the generative model improves performance on spatial reasoning due to its generation quality.
> > >
> > > Our experiments on SAT-Real and MindCube (real-world datasets) as well as SAT-Synth (synthetic) show **consistent and significant gains** when integrating world models (SWM and SVC) into the reasoning process. These results, presented in both the main paper and rebuttal, demonstrate that **with the current generation quality of SWM and SVC, our approach reliably improves performance on 3D spatial reasoning tasks**. The improvement is not tied to a single world model or dataset, but **holds across multiple architectures and domains**, providing strong empirical evidence for the effectiveness of our method.
> > >
> > > ### On the “Premature Method” Concern
> > > > While the direction is interesting, I believe it is too early to draw strong conclusions about its effectiveness for this class of tasks.
> > >
> > > Our empirical results already show that integrating controllable world models yields measurable benefits for spatial reasoning. As the **first work** to improve VLM reasoning with controllable world models in the loop, we do not claim to solve all spatial reasoning tasks. Rather, our contribution is to demonstrate the viability of this approach at the current stage of model quality, and to open a path for further research as world models rapidly improve.
> > >
> > > ### On the “zoom-in effect” interpretation
> > > > Your world model produces only minor viewpoint shifts equivalent to a zoom-in or slight reframing.
> > >
> > > **This claim is not correct.** Below we provide empirical and conceptual arguments refuting it, and invite the reviewer to cite literature if such an equivalence has been established.
> > >
> > > **Our action space is beyond “minor shifts”**
> > >
> > > As detailedly mentioned in our paper, our settings allow substantial movement:
> > > |Setting|Rotation at most|Forward at most|
> > > |-|-:|-:|
> > > |1-step|27°|0.75 m|
> > > |2-step|54°|1.5 m|
> > > |3-step|72°|2.0 m|
> > >
> > > Even the smallest setting 1-step (27° yaw, 0.75 m forward) produces parallax, occlusion change, and new line-of-sight—effects impossible to replicate with 2D zoom/crop. Larger settings yield even greater viewpoint diversity. Thus, the camera action space we investigated cannot be considered as "minor viewpoint shift".
> > >
> > > **Larger action spaces help**
> > >
> > > In both our ablation studies (Sec. 4.3) and computational cost analysis (first rebuttal), 2-step and 3-step settings consistently outperform 1-step by a large margin. If performance gains were primarily due to "minor camera shifts" equivalent to the "zoom-in effect", increasing the action space beyond small shifts should not provide further benefits. The observed trend shows the benefit comes from **3D evidence acquisition**, not 2D pixel operations.
> > >
> > > **Translation/rotation ≠ zoom/reframe**
> > >
> > > Zooming or reframing preserves projective rays and cannot reveal new surfaces visible only after translation or rotation. For example, moving forward 0.75 m can shift an object from front-left to directly left—an effect no zoom can reproduce. Many queries in our datasets rely on such view-dependent spatial changes, where 2D operations are provably insufficient.
> > >
> > > **Litrature Gap**
> > >
> > > If the reviewer maintains that small 3D motions are equivalent to zoom/reframe, we request citations defining and also **quantifying a threshold** of such equivalence.
> > >
> > > **Clarifying the role of qualitative videos**
> > >
> > > The qualitative videos in the supplementary material may misled the reviewer. The videos visualize randomly sampled camera trajectories to demonstrate controllability by comparing with ground-truth videos. They are not the trajectories used during Q&A and please refer our approach section (Sec. 3) for more information. We apologize if this caused any misunderstanding.
> > >
> > > ### Clarification on “Comparisons to Recent Works”
> > > > These methods are fundamentally designed for camera control-based video generation, aiming to improve the quality or controllability of generated videos.
> > >
> > > We would like to clarify that **our world model training shares exactly this objective**—to improve the quality and controllability of generated videos. For example, Stable Virtual Camera (SVC) is itself a camera control-based world model and serves as a world model alternative in our framework, improving performance in a zero-shot setting.
> > >
> > > As detailed in Sec. 3.2, the world model in our framework is a camera control-based video generation model, and our method requires only its ability to faithfully simulate a camera trajectory, making the “Comparison to Recent Works” section methodologically valid and directly relevant. Therefore, following our last response, our training data is sufficient for world model training with the support of recent works.

---

### Official Review · Reviewer_s68S · 2025-07-04

**Clarity:** 4
**Significance:** 3
**Originality:** 2
**Rating:** 5
**Confidence:** 3

**Summary:**

This paper introduces SpatialNavigator, which frame a Vision reasoning task with a test-time scaling framework. This framework significantly enhances the 3D spatial reasoning capabilities of many different Vision-Language Models (VLMs). The core idea is to address the inherent limitation of VLMs in understanding 3D dynamics from static 2D images by simulating a exploration of a 3D environment.

SpatialNavigator achieves this by using a VLM with a controllable video world model: the VLM iteratively proposes camera trajectories, and the world model synthesizes the corresponding egocentric views. The VLM then leverages the gathered evidences to perform spatial reasoning.

**Questions:**

- The current framework's effectiveness is demonstrated with a relatively small search depth. Beyond the computational cost, are there other fundamental considerations or architectural limitations that currently constrain scaling the Spatial Beam Search to significantly greater depths, and what are your ideas for overcoming these to enable more complex, long-horizon spatial reasoning?
- Considering the observed sensitivity of the world model's fidelity to out-of-distribution scenes and extended trajectories, what specific advancements in world model training or architecture do you believe are most critical?

**Ethical Concerns:**

["NO or VERY MINOR ethics concerns only"]

**Final Justification:**

After rebuttal and checking other reviews, I still find that it is interesting to see the world model appears orthogonal to the knowledge learned by RL-trained VLMs. This provides a good direction for future exploration and I would vote for acceptance.

**Limitations:**

- The framework currently operates within a relatively small, predefined set of primitive egocentric actions. While sufficient for the tasks presented, this limited action space raises questions about its direct generalizability to more complex, nuanced, or interactive real-world scenarios

**Quality:**

3

**Strengths And Weaknesses:**

Strengths:

- The method presents a highly intuitive approach to spatial reasoning, effectively mimicking how humans might mentally simulate movement to understand a scene from different perspectives. Starting from a static image and building a dynamic 3D understanding through simulation.
- A strength of SpatialNavigator is its design as a test-time compute framework. This makes it inherently model-agnostic. Results show that the method generally improve performance of several baseline VLM models, including strong commercial ones.
- It is particularly interesting to observe that the information provided by the world model appears orthogonal to the knowledge learned by RL-trained VLMs (e.g., "O1 style" models). This suggests that the world model provides a unique and complementary source of information

Weaknesses:
 - As a test-time compute framework, the iterative Spatial Beam Search, which involves generating multiple hypothetical trajectories and synthesizing corresponding views, incurs significant computational cost at inference time.  While this is understandable, this cost should be discussed more thoroughly, and hopefully with suggested future directions on how to improve this.

---

> ### Author Rebuttal · Authors · 2025-07-31
>
> Dear Reviewer,
>
> We appreciate the detailed comments and insightful suggestions. We are glad that you find our method both high-intuitive and effective. Here are the responses to your questions.
>
>
> ### Computational Efficiency
> We appreciate the reviewer’s suggestion and agree that analyzing computational trade-offs is important for practical deployment. To address this, we provide a structured analysis of the current computational efficiency of SpatialNavigator.
>
> Also, we would like to clarify that SpatialNavigator is the first work on test-time scaling framework that uses VLM to control a video diffusion model. Thus, we mainly focused on optimizing the effectiveness instead of efficiency, we acknowledge that efficiency optimization is a promising direction for future work. To address this, we outline several acceleration techniques that make further speed-ups highly plausible.
>
> #### Efficiency Analysis
> We profiled computational efficiency on a single H100-80G. We evaluated using gpt4o+SN(SWM), with beam size 2 and inference step 50 on SAT-Real. The memory consumption is~40G across the experiments. As SVC cam only generates high resolution video and therefore not efficient(around x3 world model latency), we mainly use SWM in this section.
> |Model|Step|SWM time(s)|vlm_search time(s)|vlm_qa time(s)|total time(s)|acc%|
> |-|-|-|-|-|-|-|
> |gpt4o|0|-|-|1.57|1.57|60.0|
> |gpt4o|1|14.52|4.70|1.69|20.91|66.0|
> |gpt4o|2|39.53|8.40|1.85|49.78|70.7|
> |gpt4o|3|52.78|13.53|3.35|69.66|68.7|
>
> In our main experiments, we focused on optimizing the effectiveness so the default setting is not the most cost-effective setting. To explore more cost-optimized settings, we further investigate reducing the number of inference step in the table below:
>
> |Iters|Step|SWM time(s)|vlm_search time(s)|vlm_qa time(s)|total time(s)|acc%|
> |-|-|-|-|-|-|-|
> |10|1|4.66|4.66|2.56|11.88|66.7|
> |10|2|7.62|8.28|2.72|18.62|66.7|
> |10|3|10.69|11.25|5.79|27.73|67.3|
> |20|1|7.22|4.44|2.48|14.14|66.0|
> |**20**|**2**|11.76|9.14|2.50|**23.40**|**68.0**|
> |20|3|23.51|10.30|4.29|38.10|67.3|
>
> The results demonstrate that our model can maintain a >95% of the accuracy gain at ≤33% of the latency, with a latency of about 23s. We tested o1 on the SAT-Real with latency in average 19.6s, so SpatialNavigator in cost-optimized setting is in the same latency band as widely used test-time-scaling pipelines.
>
> We further added the ablation study on beam size in the table below:
> |Model|Step|Beam|SWM time(s)|vlm_search time(s)|vlm_qa time(s)|total time(s)|acc%|
> |-|-|-|-|-|-|-|-|
> |gpt4o|2|1|23.24|7.51|1.60|32.35|67.3|
> |gpt4o|2|2|39.53|8.40|1.85|49.78|70.7|
> |gpt4o|3|1|32.16|14.43|3.58|50.17|68.0|
> |gpt4o|3|2|52.78|13.53|3.35|69.66|68.7|
>
> #### Scalability & Acceleration Pathways
>
> **Backbone Model**
> Recent released Wan2.1-1.3B provides videos with quality higher than our backbone CogVideoX-5B in ≈8GB VRAM. Wan2.1 achieved ≈1.7×lower (8.7s vs.14.5s) latency and ≈½ the memory footprint relative our current backbone tested under identical frame counts and resolution.
>
> **Step Distillation**
> Latent Consistency Models produce high-fidelity outputs in 2–4 inference steps, yielding 10–20× speed-ups over 50-step schedulers. Video-specific extensions such as Distribution-Matching Distillation report similar benefits on text-to-video transformers.
>
> **Low-Bit Quantization**
> Post-training methods like SVDQuant quantize both weights and activations to 4-bit while keeping FID within ±2 points of full precision. Empirical results report ≈3× speedups and up to 3.6× memory reduction on DiTs, enabling real-time sampling on a 16GB laptop-class GPU.
>
> All three techniques are open-source and multually compatible, which indicates that improving computational cost is both practical and well-supported by the community.
>
> ### Constrained scaling the Spatial Beam Search
> One fundamental constraint is error accumulation. With greater search depths, there is an increased likelihood of compounding errors in both the imagined views synthesized by the world model and the subsequent reasoning steps by the VLM. The lower fidelity of the subsequent generated views may hurt VLMs' ability to perform spatial reasoning. We further evaluated the world models' fidelity at different steps to confirm it.
>
> Using the out-of-domain simulator AI2Thor, we evaluated the world model's fidelity under a sequence of atomic actions defined in our work. We report the PSNR, LPIPS, and SSIM in Tab. 1 of the supplemental. The SVC and SWM models obtain comparable SSIM scores but the latter outperforms the former by larger margins of ~2 and 0.18 for PSNR and SSIM. We refer all reviewers to Sec. B.1.1 for more information.
> To further analyze the reliance of these world models, we evaluate the performance when extended to multiple steps(1 to 3) in the following table.
> ||PSNR$\uparrow$|SSIM$\uparrow$|LPIPS$\downarrow$|
> |-|-|-|-|
> |SVC-step1|67.38|0.997|0.39|
> |SVC-step2|63.98|0.994|0.54|
> |SVC-step3|62.80|0.993|0.57|
> |SWM-step1|67.35|0.998|0.26|
> |SWM-step2|66.71|0.997|0.30|
> |SWM-step3|65.98|0.997|0.35|
>
> Generally, we observe consistent trends across the SVC and SWM models, where their effectiveness gradually decrease as the number of steps increases. This is expected since the uncertainty of predicting the exact visual changes is compounded over steps.
>
> ### Critical advancements required
> Given our observation that the fidelity of the world model can deteriorate quickly over extended imagined trajectories, we believe that addressing this challenge is important. Some possibilities include but are not limited to a long-term memory module that is capable of storing a mental model of a 3D scene as it is explored, as well as learning to generate environments based on scene-graphs. The latter is aimed at improving generalizability to spatial layouts unseen during training time.

---

> > ### Comment · Reviewer_s68S · 2025-08-03
> >
> > Thanks for the detailed response. I believe adding the cost analysis will improve the paper. I will keep my recommendation for accpetance.

---

### Comment · Area_Chair_h3tB · 2025-08-04
**Discussion time**

Dear reviewers,

The authors appear to have put substantial work into their rebuttal. Is there anything you wish to discuss with them? Thank you s68S for engaging already. As a reminder: author-reviewer discussion will be closed on August 6 11:59pm AoE.

Best,
Your AC

---

### Note · Authors · 2025-08-12

Dear Reviewers and AC,

Thank you again for your time and valuable feedback. We are pleased that most concerns have been addressed during the discussion period. Below we summarize the remaining concerns from reviewer 27Ro and our corresponding responses.

**Concern 1 — Search World Model (SWM) training feasibility**
> It is very difficult to train a reliable world model with only 50K HM3D samples and 20K real videos.

Our training scale lies between those used in recent **camera-controlled video generation** works such as SVC, one of our world model candidates, and ReCamMaster, making it consistent with established practice.

> I do not think comparing to recent works (SVC / ReCamMaster) is appropriate. These methods are designed for camera control-based video generation, aiming to improve the quality or controllability of generated videos.

In Sec. 3.2, we specified that the world model defined in our framework is exactly a **camera controlled-based video generation model**.  During training, SWM is also optimized for the quality and controllability of generated videos. Therefore, the comparisons to recent works is appropriate to demonstrate the feasiblity of our training setting.

**Concern 2 — Empirical performance**
> The paper does not convincingly show that world models help improve spatial reasoning.

Across SAT-Real and MindCube (real-world datasets), and SAT-Synth (synthetic), integrating world models (SWM, SVC) yields consistent gains across VLM architectures. These gains hold across architectures and domains, showing that **current generation quality is already sufficient to boost spatial reasoning performance**.

Recent progress such as **Genie 3**, further reinforces the promise of our proposed direction.

**Concern 3 — “Zoom-in effect”**
> The world model produces only minor viewpoint shifts equivalent to zoom-in or reframing.

Our action space supports substantial motion (up to 72° rotation / 2.0 m translation), far beyond 2D zoom/crop capabilities. Ablations show that larger action spaces (2-step/3-step) consistently outperform smaller ones (1-step), indicating that **gains come from 3D evidence acquisition, not minor zooming/reframing**.

**Closing remark**

We believe these clarifications address the reviewer’s remaining concerns and reaffirm that our framework is both technically sound and empirically validated. We sincerely thank the reviewers and AC for their time, attention, and consideration in reviewing our additional remarks.

---

### Decision · Program_Chairs · 2025-09-17

**Decision:**

Accept (poster)

**Comment:**

This paper presents a method for pairing a VLM with a world model, in a navigation task. The VLM estimates a path to take, and the world model synthesizes the views that would be received after these actions, and these views are then sent back to the VLM as hallucinated "evidence" for scoring. This paper received mixed reviews: 5, 2, 5, 4. The positive reviews commend the intuitiveness of the approach, the fact that it's model-agnostic, and the clarity of the results. The reviews also point out (as a negative) that this approach is very expensive. Reviewer 27Ro in particular makes a confident negative case, pointing out (in addition to the speed issue) that the results are highly sensitive to the quality of the world model, and that the datasets used in evaluation are not the best ones. Overall however, the AC finds the positive case for this paper compelling, especially thanks to the additional results added in the rebuttal (MindCube dataset, GPT4.1 baseline). Therefore, the AC recommends acceptance. The authors are highly encouraged to carefully incorporate the work done during the rebuttal, to strengthen the paper for camera-ready.